



# EEAR-Clim: A high density observational dataset of daily precipitation and air temperature for the Extended European Alpine Region

Giulio Bongiovanni[1,2], Michael Matiu[2], Alice Crespi[3], Anna Napoli[2,4], Bruno Majone[2], and Dino Zardi[2]

[1]University School for Advanced Studies Pavia (IUSS), Pavia, Italy
[2]University of Trento, Trento, Italy
[3]Center for Climate Change and Transformation, Eurac Research, Bolzano, Italy
[4]Center for Agriculture Food Environment (C3A), Trento, Italy

**Correspondence:** Giulio Bongiovanni (giulio.bongiovanni@unitn.it)

**Abstract.** The Extended European Alpine Region (EEAR) exhibits a well-established and very high-density network of in-situ weather stations, hardly attainable in other mountainous regions of the world. However, the strong fragmentation into national and regional administrations and the diversity of data sources have so far hampered full exploitation of the available data for climate research. Here, we present EEAR-Clim, a new observational dataset gathering in-situ daily measurements of air temperature and precipitation from a variety of meteorological and hydrological services covering the whole EEAR. Data collected include time series from recordings up to 2020, the longest ones spanning up to 200 years. The overall observational network encompasses about 9000 in-situ weather stations, significantly enhancing data coverage at high elevations and achieving an average spatial density of one station per 6.8 $km^2$ over the period 1991-2020. Data collected from many sources were tested for quality to ensure internal, temporal, and spatial consistency of time series, including outliers removal. Data homogeneity was assessed through a cross-comparison of the outcomes using three methods well established in the literature, namely Climatol, ACMANT, and RH Test. Quantile matching was applied to adjust inhomogeneous periods in time series. Overall, about 4% of data were flagged as non-reliable and about 20% of air temperature time series were corrected for one or more inhomogeneous periods. In the case of precipitation time series, fewer breakpoints were detected, confirming the well-known challenge of properly identifying inhomogeneities in noisy data. The dataset aims to serve as a powerful tool for better understanding climate change over the European Alps.

## 1 Introduction

The continuous warming of the climate is amplified in mountain regions (Hock et al., 2019), and the European Alps have been found particularly vulnerable to climatic changes (Cramer et al., 2020). Projected future changes in Alpine climate envisage rising temperatures, changes in the seasonal cycle of precipitation and runoff, increasing frequency of temperature and precipitation extremes, snow cover reduction and glacier shrinking (Gobiet et al., 2014). The assessment of climate change in the Alpine region relies on the analysis of climate observations (Hartmann et al., 2013) and benefits from a density of weather stations and length of data series not easily attainable in many other regions (Brunetti et al., 2009). However, the fragmentation





of their owners and the diversity of data sources make collecting and managing such datasets a rather complex task (Auer
et al., 2007; Andrighetti et al., 2009; Chimani et al., 2023). Indeed, many studies regarding the Alpine region were hindered
by a scarcity of data sharing, harmonized data portals, and joint projects or initiatives fostering such analyses (Beniston et al.,
2018).

To overcome these limitations, several datasets collecting meteorological observations have been developed in recent decades
in Europe. Among these, one of the most widely used is E-OBS (Klein Tank et al., 2002; Cornes et al., 2018), a daily gridded
observational dataset based on the European Climate Assessment and Dataset (ECA&D) database of meteorological mea-
surements for precipitation, air temperature, relative humidity, sea level pressure, global radiation and wind speed in Europe.
However, E-OBS is known to be affected by significant biases in some areas, such as the Southern part of the Alps, where
the ECA&D database has a density of stations compared to other regions (Hofstra et al., 2009; Kyselý and Plavcová, 2010).
Likewise, the dataset HISTALP (Historical Instrumental Climatological Surface Time Series Of The Greater Alpine Region:
Auer et al. (2005)) has the advantage of gathering measurements of different climate variables and focusing specifically on
the Alpine region. The primary goal of HISTALP was to achieve long-term temporal consistency. Accordingly, as long-term
time series are rare, HISTALP spatial density remains quite low compared to what is needed to reproduce the strong spatial
variability associated with the complex nature of Alpine terrain (Eccel et al., 2012). Most of the best spatially resolved datasets
are organized on a national basis (Herzog and Müller-Westermeier, 1996; Brunetti et al., 2001; Lussana et al., 2019), hence
they are confined by national borders (Auer et al., 2005). More recently, the Alpine Precipitation Grid Dataset APGD (APGD:
Isotta et al. (2014)), covering up to 2019 in its recently updated version, was developed for the Alpine region. APGD is based
on the extended network of rain-gauges available over the Alpine region and significantly improves the spatial density of
HISTALP stations, reaching an average of one station every 10 km, but it covers only precipitation. The Iberia dataset (Her-
rera et al., 2019), covering the Iberian Peninsula, including the Pyrenees, exhibits a spatial density comparable to APGD. For
other regions of the world, datasets including larger amounts of collected time series can be found, but mostly covering larger
areas, and thus attaining lower spatial resolutions, both for national and supranational products (Yatagai et al., 2012; Livneh
et al., 2015; Cesar Aybar and Felipe-Obando, 2020; Tang et al., 2020; Daly et al., 2021; Hatono et al., 2022; Han et al., 2023).
Multi-parameter datasets such as E-OBS and HISTALP are essential because they allow the detection of changes in the regimes
of the different variables, leading to increased confidence in the results from climate studies (Brunetti et al., 2009). Indeed,
the simultaneous analysis of a wide spectrum of meteorological variables allows a better understanding of the atmospheric
processes that modulate and trigger the variability and trends shown by the single meteorological parameters, and the mutual
interactions linking the different variables (Gaffen and Ross, 1999; Kaiser, 2000; Wang and Gaffen, 2001; Huth and Pokorná,
2005; Beniston, 2006). Time resolution is another important issue when studying climate change. Compared to the past, recent
climatological research is even more focused on the identification of changes in the frequency and intensity of extreme weather
events, which require datasets with at least daily resolution (Jones et al., 1999; Folland et al., 2000). Furthermore, daily data
are mandatory in the model applications for simulation of bio-ecological, agricultural, and hydro-climate systems (Eccel et al.,
2012).



Hydro-climate modelling, as well as model evaluation, use data from meteorological observations as forcings to provide a representation as close as possible to real environmental conditions. However, data quality may strongly impact climate and hydrological studies results and predictions in terms of reliability, accuracy and precision (Laiti et al., 2018). For instance,
accurate observational data are needed to improve the correction of possible biases in the model output. The reliable analysis of the evolution of key climate variables plays an important role in the current discussion on climate change (Begert et al., 2005). Stakeholders also require data of high quality and representativeness (Ha-Duong et al., 2007; Swart et al., 2009) to prevent and quickly plan for disaster management, risk mitigation, and elaborate proper local adaptation strategies. Therefore, there is a clear need for high-quality observational datasets to deepen and improve our knowledge about climate, its change,
and variability (Skrynyk et al., 2023).

The process of recording, collecting, digitising, processing, transferring, storing, and transmitting climate data series may introduce many errors affecting data quality (Brunetti et al., 2006). A variety of data quality issues, such as shifting in units and time frequency of measurements, malfunctioning of sensors, erroneous data recording, transcription, or processing, are addressed by a specific process called Quality Control (QC) (Fiebrich and Crawford, 2001; WMO, 2017). In addition, non-
climatic factors may introduce discontinuities in recorded time series, such as changes in measuring methods, units or instruments, calculation methods, ambient modifications, as well as station relocation or maintenance (Alexandersson and Moberg, 1997; Peterson et al., 1998; Aguilar et al., 2003; Auer et al., 2005; Venema et al., 2013; Gubler et al., 2017). Such discontinuities, or inhomogeneities, give rise to biases in datasets, leading to misinterpretations of the climate patterns and, thus, inaccurate or even wrong interpretations of trends and climatologies. Therefore, such inhomogeneities have to be detected and
removed by means of suitable homogenization procedures (Peterson et al., 1998; Aguilar et al., 2003; Trewin, 2010; Begert et al., 2005). Quality Control (QC) and homogenization procedures can be applied on time series of various climate variables with either monthly or daily or hourly time resolution (Trewin, 2013; Fioravanti et al., 2019; Squintu et al., 2019; Mateus and Potito, 2021; Dijkstra et al., 2022).

Depending on specific goals and approaches, different existing QC methods can be used (Faybishenko et al., 2022). QC is
often performed semi-automatically (Hubbard et al., 2005), or automatically on large amounts of data. However, despite its practical convenience, automated QC may fail, resulting in erroneously flagging good observations as invalid (Fiebrich and Crawford, 2001). The detection of outliers is the phase of the QC most prone to this type of error (Kuhn and Johnson, 2013). Comprehensive reviews of the many homogenization methods developed over time to detect inhomogeneities can be found in Peterson et al. (1998); Aguilar et al. (2003); Reeves et al. (2007); Ribeiro et al. (2016). To date, the development and use of
homogenization methods focused mainly on temperature and precipitation time series and monthly rather than daily time-scale (Thorne et al., 2011; Venema et al., 2012). Inhomogeneities detected during the homogenization process are ideally identified and confirmed from the analysis of metadata containing details of the station's history. However, this is rarely possible because usually metadata have not been digitized, or they were not recorded at all (Brugnara et al., 2023; Guijarro et al., 2023). A highly recommended approach, ensuring higher confidence in breakpoints detection, consists of a combination of different methods
and inter-comparison of their results (Brunetti et al., 2006; Toreti et al., 2012; Kuglitsch et al., 2012; Ribeiro et al., 2016).
Given the challenges posed by observational datasets available for the Alpine region, it becomes evident that overcoming these

limitations is crucial to improve our understanding of how climate change is affecting the area. Hence, the overall objective of the present study is to develop a new and unprecedented observational dataset for the European Alpine region, addressing key issues such as data quality, spatial density, time resolution, and completeness. In particular, QC and homogenization procedures

are applied to station time series, both combining already existing methods and developing new ones.

The paper is structured as follows. In section 2, the study domain is framed from a geographical and climatic point of view, and the collected data are described in terms of their distribution in space, time, and elevation. Section 3 presents data QC, and section 4 presents the homogeneity assessment of the time series. The last two sections, 5 and 6, are dedicated to the discussion of the results and conclusions.

## 2   Study Area and Data Collection

### 2.1   Study Area

The EEAR-Clim dataset includes observations from a very dense network of in-situ weather stations located within the Extended European Alpine Region (EEAR), i. e. the region shown in fig. 1, lying between 3°E and 18°E in longitude, 43°N and 49°N in latitude. The domain covers an area of about 800,000 $km^2$, extending over 1100 km from Central France to Western

Hungary in the West-East direction and over 700 km from South Germany to Central Italy in the North-South direction. The domain includes the entire territories of Switzerland, Liechtenstein, Austria and Slovenia, as well as parts of France, Italy, Germany, Croatia, Czech Republic, Slovakia, Hungary, Bosnia and Herzegovina.

The EEAR is predominantly constituted by complex terrain and hence characterized by strong elevation gradients, with terrain heights ranging from -5 m above sea level (m a.s.l.) at San Giuseppe di Comacchio (Italy), to the top of the Alps, 4807 m a.s.l. at the Mont Blanc summit (Italy-France). In particular, the EEAR is centered on the European Alps, an arc-shaped

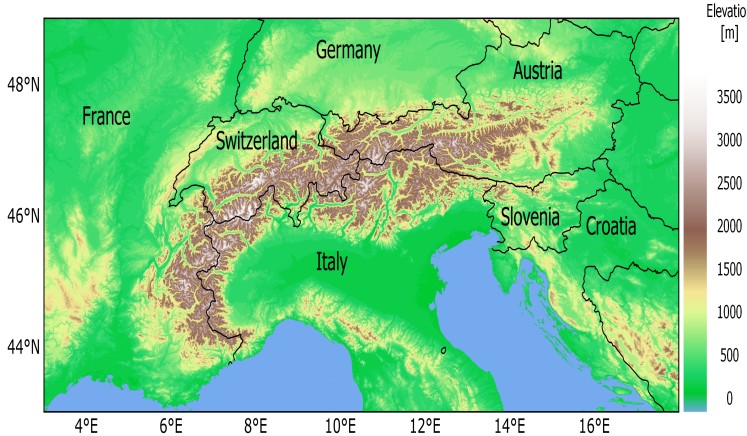

**Figure 1.** Overview of the Extended European Alpine Region (EEAR). The orography is based on the Copernicus Digital Elevation Model EEA-10 (https://spacedata.copernicus.eu/collections/copernicus-digital-elevation-model).



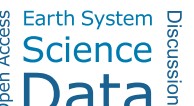

mountain range extending about 1300 km, delimited to the West by the Bocchetta di Altare (459 m a.s.l.), in northern Italy, and to the East by the Godovič Pass (850 m a.s.l.), in Slovenia. Several sub-alpine mountain ranges surround the Alps, including the Jura mountains and the Massif Central to the West, the Black Forest and the Bohemian Forest to the North, the Dinaric Alps to the East and the Apennines to the South. The Alps, one of the major mountain ranges in Europe, are characterized by diverse climate features influenced by several large-scale weather regimes (Schär et al., 1998; Auer et al., 2005; Panziera et al., 2015).

Moreover, the complex topography of the Alps and the surrounding mountain chains induce several additional effects on the local climate. These include orographic lifting and the related rain-shadow effect, air channelling and blocking contributing to phenomena such as föhn winds or thermally driven orographic winds(Serafin and Zardi, 2011; Laiti et al., 2014; Giovannini et al., 2017), influence on temperature patterns through elevation gradients (Auer et al., 2007; Marchetti et al., 2017).

Also, the southern part of the region stretches into the Mediterranean Sea, another area identified as a climate change hotspot

(Hartmann et al., 2013). The presence of these two climate change hotspots, namely the Alps and the Mediterranean Sea, enhances the vulnerability of the region to climate impacts and further motivates the analysis of climatic patterns from observations.

## 2.2   Data Collection

The EEAR-Clim dataset presented here includes time series of daily mean, maximum and minimum air temperature (indicated

respectively as $T$, $T_{max}$, and $T_{min}$) and total precipitation ($TP$). Data were collected from different global, national, regional and local providers across the EEAR. Table 1 summarizes the data providers and the number of stations for each country of the region, as well as the amount of time series for each variable and the time period of observations. Most of the data are distributed by national providers, except in Italy where meteorological stations are operated by local and regional institutions. Bosnia and Herzegovina faces challenges in the availability of daily climate series due to historical issues related to the dissolution of

the former Yugoslavia (Auer et al., 2005). However, a few time series from that country were obtained through the Global Historical Climatology Network (GHCN) (Vose et al., 1995). Fig. 2a shows the time evolution of the available time series through 10-year increments up to 2020. The longest records date back to the mid-$18^{th}$ century, though these are limited to a few stations. About 50% of the available time series cover a 30-year timespan, long enough to capture key climatological features. Generally, the availability of time series rapidly decays for periods longer than 60 years for air temperature, and 90

years for precipitation. Fig. 2b instead shows the overall distribution of available time series during the entire observation period, highlighting precipitation as the variable with the largest number of available stations. The sudden increase in station availability from the early 1990s is due to the increasing deployment of new automatic weather stations (WMO, 2008) and the missing digitization of pre-1990 records.

Fig. 3a depicts the spatial distribution of stations measuring at least one variable, highlighting the amount of nearest neigh-

bours within 10 km radius and their density vs. elevation. The distance between stations is a useful metrics to assess the network density. The spatial density of weather stations is highly variable during the whole period. However, during 1991-2020 the average density is about one station every 6.8 $km^2$, the highest values being reached by observational datasets over the Alpine region. Switzerland, the Northern Apennines, and the main Alpine range are characterized by the highest density of stations,



**Table 1.** Overview table of available stations for each data provider and variable. Columns "STARTING YEAR" and "OPEN DATA" respectively report the starting year of the series, and whether data are available without restrictions, upon provider's policy.

| COUNTRY | PROVIDER | TOTAL | T | Tmax | Tmin | TP | STARTING YEAR | OPEN DATA |
|---|---|---|---|---|---|---|---|---|
| *Austria* | Bundesanstalt für Geologie, Geophysik, Klimatologie und Meteorologie (Geosphere) | 505 | 496 | 495 | 495 | 475 | 1852 | yes |
| | Hydrographische Archivdaten Österreichs (eHYD) | 1021 | 607 | 0 | 0 | 881 | 1969 | yes |
| *Bosnia Herzegovina* | Global Historical Climatology Network (GHCN) | 2 | 2 | 2 | 2 | 2 | 2001 | yes |
| *Croatia* | Državni HidroMeteorološki Zavod (DHMZ) | 18 | 18 | 18 | 18 | 18 | 1857 | yes |
| *Czech Rep.* | Český HydroMeteorologický Ústav (CHMU) | 72 | 13 | 11 | 11 | 71 | 1961 | yes |
| *France* | Météo-France | 1120 | 868 | 888 | 888 | 803 | 1922 | no |
| *Germany* | Deutscher WetterDienst (DWD) | 1074 | 251 | 245 | 245 | 1056 | 1781 | yes |
| *Hungary* | Országos Meteorológiai SZolgálat (OMSZ) | 199 | 39 | 39 | 39 | 186 | 1901 | yes |
| *Italy* | Agenzia Regionale per la Protenzione dell'Ambiente del Friuli Venezia Giulia (ARPA FVG) | 188 | 170 | 170 | 171 | 178 | 1991 | yes |
| | Agenzia Regionale per la Protenzione dell'Ambiente della Lombardia (ARPA Lombardia) | 450 | 332 | 337 | 336 | 438 | 1763 | yes |
| | Agenzia Regionale per la Protezione Ambientale del Piemonte (ARPA Piemonte) | 323 | 303 | 303 | 303 | 314 | 1913 | yes |
| | Agenzia Regionale per la Prevenzione, l'Ambiente e l'Energia dell'Emilia-Romagna (ARPAE) | 519 | 367 | 368 | 367 | 467 | 1961 | yes |
| | Agenzia Regionale per la Protezione dell'Ambiente Ligure (ARPAL) | 201 | 186 | 169 | 169 | 193 | 2002 | yes |
| | Agenzia Regionale per la Prevenzione e protezione Ambientale del Veneto (ARPAV) | 303 | 245 | 245 | 245 | 268 | 1956 | no |
| | European Climate Assessment & Climate (ECA&D) | 63 | 10 | 10 | 10 | 62 | 1813 | yes |
| | Fondazione Edmund Mach | 9 | 9 | 9 | 9 | 8 | 1983 | no |
| | Meteo Aeronautica Militare (Meteo AM) | 24 | 20 | 20 | 20 | 20 | 1813 | yes |
| | MeteoTrentino | 181 | 176 | 175 | 175 | 159 | 1920 | yes |
| | Provincia Autonoma di Bolzano | 218 | 189 | 187 | 187 | 99 | 1920 | yes |
| | Regione Marche | 113 | 79 | 79 | 79 | 50 | 1951 | yes |
| | Regione Toscana | 310 | 180 | 180 | 180 | 305 | 1916 | yes |
| | Regione Umbria | 42 | 35 | 35 | 35 | 40 | 1916 | yes |
| | Regione Autonoma Valle d'Aosta | 78 | 78 | 78 | 78 | 72 | 1866 | yes |
| *Slovakia* | Slovenský HydroMeteorologický Ústav (SHMU) | 103 | 17 | 17 | 17 | 98 | 1991 | yes |
| *Slovenia* | Agencija Republike Slovenije za Okolje (ARSO) | 467 | 167 | 172 | 172 | 457 | 1960 | yes |
| *Switzerland* | MeteoSwiss | 1329 | 629 | 586 | 594 | 1149 | 1863 | no |
| *EEAR* | | *8932* | *5486* | *4838* | *4845* | *7869* | | |

with each having at least 10 neighboring stations within a 10 km radius. This high density is more appreciable when compared

to other areas, such as the French pre-Alps or the Po Valley. However, a minimum density of at least 5 neighbouring stations within a 10 km radius includes 70% of the stations across the whole EEAR. The southeastern part of the domain (Croatia) is the area characterized by the lowest density of stations. This is due to restrictions on local data providers and missing daily measurements. Fig. 3b shows the number of stations by elevation and the respective covered area, considering all stations with at least one variable measured and 100-m elevation ranges. More than 50% are located below 500 m a.s.l., while about 10% of

them are above 1500 m a.s.l. Despite the varying distribution with elevation, the density of stations per area in these elevation ranges is comparable, which is in line with the average density over the EEAR. Fig. 4 shows the spatial distribution of the stations in the EEAR, highlighting for each station the time coverage, in years, over the measurement period. Clearly, it confirms previous considerations in terms of inter-stations distance (fig. 3a), but, in addition, it highlights some aspects typical of each variable. It is evident the higher spatial resolution of the rain-gauge network, as well as the longer time extent of precipitation

time series. Air temperature stations show a lower density, particularly in the area surrounding the Alps, such as Germany and

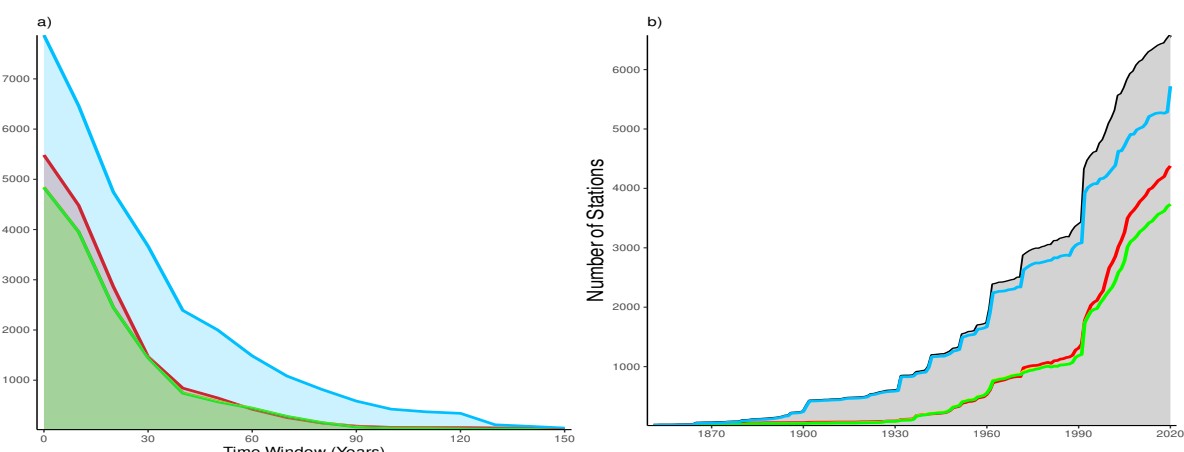

**Figure 2.** Distribution of stations by time series length (a) and time (b). Coloured lines identify each variable:, i.e. mean (in red), maximum and minimum (in green) air temperature, and precipitation (in light blue). In b) the gray shaded area shows the total number of stations with at least one measured variable in the year.

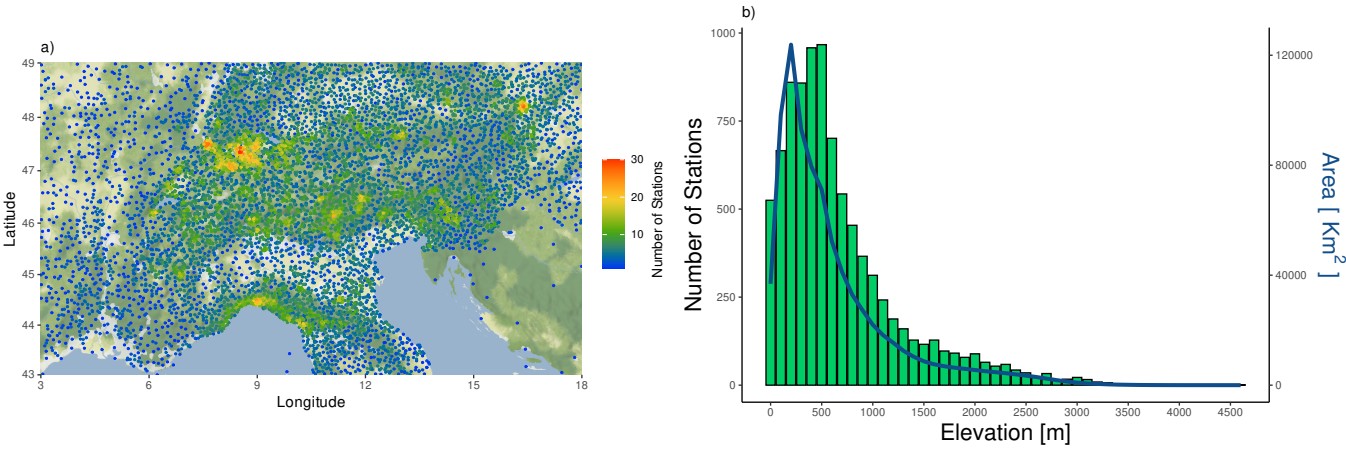

**Figure 3.** Horizontal and vertical distribution of stations within the EEAR. In a) for each station with at least one measured variable, the color scale highlights the number of nearest neighbours within a ten km radius. In b) green bars show the number of stations in each 100-m elevation band, while the blue line represents the respective area covered, based on EU-DEM 1.1.



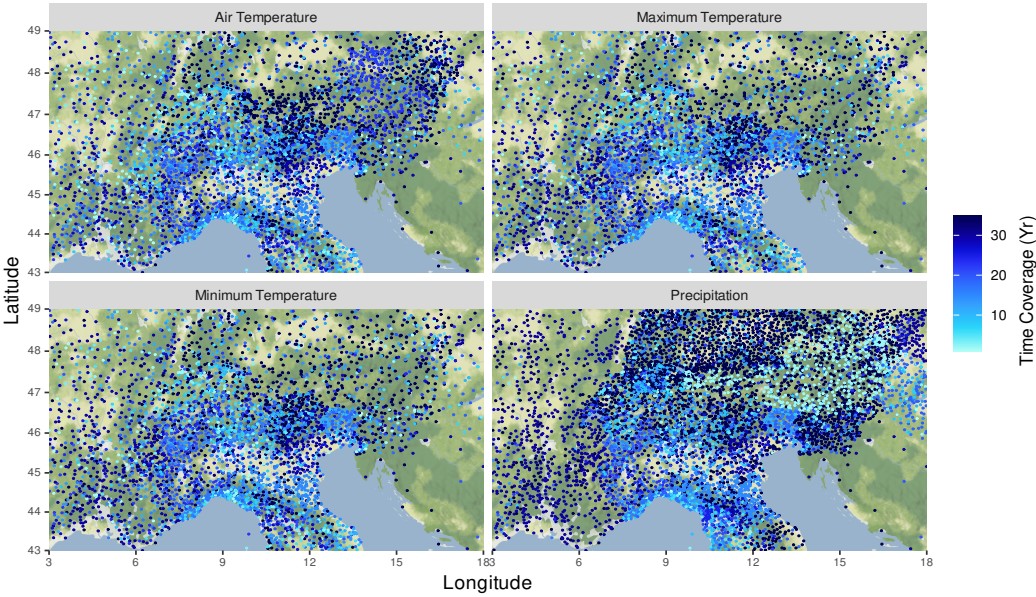

**Figure 4.** Stations distribution for mean, minimum and maximum air temperature, and precipitation. The colorbar indicates the length of the time series expressed in years. Time series exceeding 30 years are shown with the same color (the darkest blue).

Slovenia. Moreover, a different coverage of Austria among the air temperature variables time series is due to the availability of only mean temperature measurements for eHYD data provider.

## 3 Methods

The twofold QC-homogenization process adopted here involves several steps, illustrated by the flow-chart in fig. 5 and explained in detail in the subsections indicated therein.

### 3.1 Data Processing

Data collected from different sources undergo preliminary inspection to identify and address potential issues related to measurement, recording, digitization, transmission, and processing. Data from each source come with their own format and peculiarities; hence, initial standardization is essential. Accordingly, data are first converted into a common format, ensuring consistency across the dataset. Proper labelling of missing values is verified by comparing them with quality codes in the metadata when available. Daily values are computed for time series provided at sub-daily temporal resolutions. Data are subsequently checked for duplicate time stamps and missing dates. Time series shorter than one year or without valid data are removed. This pre-processing phase is useful as preliminary screening before QC procedures. After this stage, stations meta-

Earth System
Science
Data

**Figure 5.** Flow-chart of QC and homogenization procedures. Numbers in brackets represent the subsections in which the corresponding method is presented.

data are merged into comma-separated value (.csv) files, one for each variable. Each file includes information about the station

name, latitude, longitude, elevation, data provider, country and a unique alphanumeric code identifying each station.

## 3.2 Intra-stations Quality Control

Quality control within time series aims at assessing internal and temporal consistency of time series, following the criteria suggested by the World Meteorological Organization (WMO) (WMO, 2017, 2018), and integrating methods proposed by various authors (Cerlini et al., 2020; Crespi et al., 2018; Curci et al., 2021; Durre et al., 2010; Faybishenko et al., 2022;

Fioravanti et al., 2016, 2019; Isotta et al., 2014; Matiu et al., 2021) with new add-ons. The selection and application of these methods depend on the specific variable and its statistical distribution. A summary of intra-stations QC tests is reported in

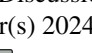



table 2. The selected tests are run independently and automatically, generating flags for each observation to highlight anomalous

**Table 2.** Resume table of all the main tests applied to check intra-stations consistency.

| TEST | DESCRIPTION | PARAMETERS |
|---|---|---|
| ***Time consistency*** | | |
| Repeated values | Repeated values for 5 or more consecutive days | $T, T_{min}, T_{max}, TP$ |
| Repeated zeros | Repeated zero precipitation values for 6 or more consecutive months | $TP$ |
| Time step | $|X_t - X_{t-1}| \leq 20°C$ | $T, T_{min}, T_{max}$ |
| ***Internal consistency*** | | |
| Range check | $-50°C \leq X \leq 50°C$ | $T, T_{min}, T_{max}$ |
| | $0mm \leq X \leq 500mm$ | $TP$ |
| First consistency test | $T_{min} \leq T \leq T_{max}$ | $T, T_{min}, T_{max}$ |
| Second consistency test | $0°C < (T_{max} - T_{min}) < 30°C$ | $T_{min}, T_{max}$ |
| ***Outliers detection*** | | |
| MAD method | $SDO = \left| \dfrac{X - median(X)}{median(|X - median(X)|)} \right| > 3$ | $T, T_{min}, T_{max}$ |
| Percentile-based method | $TP > 9_{p_{95}} \text{ if } T \geq 0 \mid \nexists T$ | $TP$ |
| | $TP > 5_{p_{95}} \text{ if } T < 0$ | $TP$ |

values saved in log files. Abnormal values are manually inspected in a conservative way, i.e. flagging as missing only the values that are definitely erroneous and avoiding the removal of valid observations. The time consistency check examines the rate of
change of data over time through two tests. The repeated values test inspects sequences of identical readings prolonged for more than five days, which is crucial for identifying data entry errors. In the case of precipitation data, extended sequences of 0 mm, possibly due to erroneous transcription of missing data (Peterson et al., 1998), are identified and replaced with missing value flags if they exceed 180 days. In addition, a step check is applied to air temperature data, comparing consecutive temporal changes to the step limit value of 20°C, equal to the maximum permitted day-by-day variation. This ensures that
sudden, unrealistic jumps in temperature readings are flagged and reviewed for data quality issues.

The internal consistency tests aim to identify major errors in time series from inspections of data within reasonable ranges. In particular, the range check evaluates whether daily measurements fall within physically consistent ranges based on historical records (WMO, 2017). Air temperature data are validated against extreme values of -50°C, close to the minimum record of -49.6°C on 10 February 2013 in Busa Fradusta (Pale di San Martino, Italy), and 50°C, which includes the highest record
of 45.9°C on 28 June 2019 in Gallargues-le-Montueux (France). Similarly, the highest record of 948.4 mm recorded on 7th



October 1970 in Genoa Bolzaneto (Italy), is considered in setting precipitation thresholds. However, because of uncertainties related to precipitation measurements, conservative thresholds of 0 and 500 mm are set.

Consistency check tests assess the relationship between two or more parameters, comparing observations to evaluate physical and climatological consistency (WMO, 2017). Specifically, two consistency checks compare mean, maximum and minimum air temperature. One evaluates whether the mean temperature falls between its minimum and maximum daily values. The other method focuses on the difference between minimum and maximum temperatures, evaluating whether it is non-zero and within a given threshold, set at 30°C, to capture realistic temperature variations. Data corruption or measurement errors can produce outliers, i.e. observations that significantly deviate from the others (Aggarwal, 2017; Hawkins, 1980). Outlier detection is aimed at identifying statistical anomalies within the distribution of time series values, and it is the test that requires the most careful attention. Thus, an ex-post manual verification is always recommended.

In the literature, mean and standard deviation are conventionally used to detect outliers, assuming a normal distribution of data. However, the presence of outliers and skewed data distributions can compromise the effectiveness of these two metrics. A typical solution to address the issue of skewness is data symmetrization, such as the application of a Box-Cox transformation (Rayens and Srinivasan, 1991), although this method may not reliably identify outliers. A more robust alternative for outliers detection is the use of the median and the median absolute deviation (MAD). Indeed, the median is less sensitive to the presence of outliers (Leys et al., 2013; Hunziker et al., 2018), and it can be easily adapted to skewed distributions without losing robustness (Meropi et al., 2018). In this study, outliers of air temperature data are detected using median and MAD, as suggested by Leys et al. (2013). The Stahel-Donoho outlyingness $SDO$ (Pavlidou and Zioutas, 2014) is adopted: an outlier is detected if the $SDO$ value exceeds a predefined threshold of 3, according to a conservative outliers removal (Miller, 1991). In the case of precipitation data, detection methods often rely on upper percentile-based thresholds (Cerlini et al., 2020). Here, we consider two different thresholds, respectively 5 and 9 times the $95^{th}$ percentile, contingent upon the availability of precipitation data for the tested time series.

### 3.3 Study of Spatial Consistency

After intra-stations QC, the resulting time series undergo fully automatic spatial consistency tests, and data flagged with warning flags are automatically replaced with missing values. The spatial consistency tests are crucial as they identify further inconsistencies that were not detected by previous checks. The tests compare the records of each time series, called target stations, with those of nearest neighbours, called reference stations. However, when daily time series of different stations are compared, the issue of time shifting may arise. Indeed, observational times may differ among different stations and data providers, especially for precipitation data, hindering the comparability of daily records (Schmidlin et al., 1995; Kunkel et al., 2005; Reek et al., 1992). The time shifting issue is faced by a three-day moving window comparison, i.e. each daily value in target station is compared to those of neighbouring stations over a three-day window centered on the test day.

Before the application of the tests, candidate reference stations are selected based on specific criteria outlined in table 3. Only stations within a 50-km radius around the target station are considered. In the case of air temperature time series, stations





**Table 3.** Overview of parameters used to select reference time series for spatial QC and breakpoints detection. Geographic distance, in km, between station points is computed by the R package geosphere (Hijmans et al., 2021). The elevation difference parameter, expressed in m, is used to reject candidate stations located at elevations too different from the target station. The number of surrounding reference stations defines the lower and upper limits of candidate stations that can be selected. Values on parenthesis show specific thresholds used for precipitation data. Values labelled as "N.A." refer to parameters not considered when selecting reference time series.

| PARAMETER | QUALITY CONTROL | HOMOGENIZATION |
|---|---|---|
| *Distance [km]* | 50 | 100 |
| *Elevation difference [m]* | 100 (N.A.) | 300 |
| *Pearson's correlation coefficient* | 0.8 | 0.9 (0.8) |
| *Valid data [%]* | 80 | 70 |
| *Time length [Yr]* | N.A. | 30 |
| *Number of surrounding reference stations* | 3 - 10 | 4 - 25 |

with an absolute elevation difference exceeding 100 m are rejected. Further selection criteria include a Pearson's correlation coefficient threshold of 0.8 and a maximum allowable missing data percentage of $20\%$ over the common period with the target station (Alexander et al., 2006; Toreti and Desiato, 2008). Typically, a set of a minimum number of 3 and a maximum of 10 reference stations is identified for each target station. When a set includes more, only the closest 10 are retained. Conversely, if fewer than three candidates meet the criteria, no test is applied.

### 3.3.1 Wet and Dry Isolated Reports Test

This test is applied solely to precipitation time series, following Isotta et al. (2014). The main goal is to assess whether wetness or dryness daily conditions observed at the target station are corroborated by the reference time series. The distinction between wetness and dryness depends on whether the total precipitation exceeds a given threshold, defined as in Isotta et al. (2014). Wetness conditions at the target station are defined when the daily precipitation amount exceeds a threshold depending on the distance between the target and the closest reference station, as well as the period of the year. The threshold is computed as follows:

$$th_{tp} = f_{wd} + f_{min} \frac{d_{min}}{d_{th}} \tag{1}$$

where $d_{min}$ is the distance between the target and reference station, $d_{th}$ is the distance threshold, here set to 15 km (Isotta et al., 2014), $f_{min}$ and $f_{wd}$ are constants, both expressed in mm. The test is applied only if $d_{min}$ does not exceed a tolerance value equal to $d_{th}$. However, during the convective season, from May to September, the tolerance value, but not $d_{th}$, is increased to 20 km to account for the higher variability associated with that season, although $d_{th}$ remains unchanged. The constants $f_{min}$ and $f_{wd}$ are 3.2 mm and 0.3 mm during the convective period and 2.7 mm and 0.3 mm otherwise. The test confirms wetness conditions at the target station if at least one reference station records a precipitation amount higher than 0.3 mm within a three-day moving window centered on the tested day.



The procedure for testing dryness conditions mirrors the wetness case but with reversed thresholds. According to Isotta et al. (2014), dry conditions at the target station are defined when the daily precipitation amount is below 0.3 mm. For the reference stations, the threshold is computed using eq. (1), but with $f_{wd}$ increased to 0.8 mm. The test confirms dryness conditions at the target station if, within the same three-day moving window, at least one reference station records a precipitation amount lower than 0.3 mm. When isolated dry or wet conditions are detected, the respective values at the target station are flagged as missing data.

### 3.3.2 Anomaly-based Tests

Anomalies-based tests focus on climatological anomalies, i.e. deviations of observed data from their long-term average. These tests are applied to the time series of all the variables with at least 30 years of data. The selected reference time series are limited to the time extent of the target station. The resulting set of time series is used to compute a daily climatology by averaging the values over all years and using a moving window centered on the considered day. The window length depends on the variable considered. In the case of air temperature, a 15-day moving window is used, while for precipitation, zero values are excluded, and the window length is increased to 30 days. The daily climatology is computed using $ts2clm$ function from the R package $heatwaveR$ (Schlegel and Smit, 2021). Finally, daily anomalies from climatologies are computed for each target station.

The first test, known as the *corroboration method*, follows Durre et al. (2010) and Curci et al. (2021). Anomalies at the target station are compared to those at reference stations using a 3-day moving window centered on each day, assessing whether for at least one reference station the discrepancy is below a given threshold. The threshold is determined through sensitivity tests on raw time series, aimed at finding the optimal value that allows both the successful detection of previously identified outliers and the reduction of false outliers flagging. The selected value is set at $10°C$ for air temperature and 50 mm for precipitation. If the target station anomaly is not corroborated by any reference time series anomalies, the daily value is flagged as an outlier. An additional control for precipitation data involves computing the relative difference between target and reference time series anomalies. If the relative error exceeds $50\%$, the suspicious anomalous values are labeled as outliers.

The second test, based on methods suggested by Matiu et al. (2021) and Crespi et al. (2018), reconstructs target station values averaging the quantities $x_{r,j}$ computed for each reference station $j$:

$$x_{r,j} = x' + y_{anom,j} - x_{anom} \tag{2}$$

where $x_{anom}$ and $y_{anom,j}$ are the anomalies of the target and reference station $j$, respectively, computed following the same procedure as the corroboration method. Here, $x'$ denotes the target station time series with missing values reconstructed from neighbouring stations using a spatial interpolation approach:

$$x'_i = \begin{cases} \frac{\sum_j w_j (x_{i,j} + \cdot cf_j)}{\sum_j w_j} & \text{if } x_i = NA \\ x_i & \text{otherwise} \end{cases} \tag{3}$$





where $x_{i,j}$ are the data from reference time series $j$ for day $i$, and $w_j$ are the weights defined as:

$$w_j = e^{-\frac{\left(1-r_{ij}{}^2\right)}{\frac{\tau_r^2}{log2}}} \tag{4}$$

with $r_{ij}$ being the correlation coefficient between the target $i$ and reference station $j$, and $\tau_r$ a constant equal to 0.3. In eq. 3, $cf_j$ is the correction factor, relating the target and reference series based on their climatological conditions. For precipitation, $cf_j$ is the ratio of the averages of daily data between the target (excluding the daily record under reconstruction) and the $j$th reference time series. For air temperature, $cf_j$ is the difference between these averages. The correction term $x_{i,j} + \cdot cf_j$ means $x_{i,j} \cdot cf_j$ for precipitation, and $x_{i,j} + cf_j$ for air temperature. Finally, the original $x$ and reconstructed $x_{r,j}$ time series are compared using the same twofold procedure as the corroboration test. This includes the application of the 3-day moving window comparison, and, for precipitation data, the assessment of whether the relative difference between $x$ and $x_{r,j}$ is below 50%.

## 3.4 Break Detection Methods

The high density of the dataset allows for a robust assessment of time series homogeneity. However, dealing with a large number of time series, homogenization has to be carried out by selected automatic methods. Although an unsupervised homogenization procedure is not recommended (Aguilar et al., 2003), its implementation is now quite common (Ribeiro et al., 2016), because most methods exploit iterative inter-comparisons of several nearby and correlated stations (Curci et al., 2021). The application and comparison of different algorithms is strongly suggested (Toreti et al., 2011; Kuglitsch et al., 2012; Ribeiro et al., 2016; Brugnara et al., 2023), particularly if station metadata are not available. This approach reduces false break detection and increases confidence in accepting or rejecting breakpoints. Another issue related to dataset homogenization is the high amount of computational resources required. In this respect, the best solution is to run break detection methods on single stations to optimize the process - i.e., to reduce the computational load and increase the reliability of detected inhomogeneities.

The selected methods for break detection have to be accurate and reliable in detecting breakpoints, permit their execution in automatic mode, given the large amount of stations, tolerate missing values without limitations, and allow homogenization of time series without restrictions in time or quantity.

After careful comparisons among the various options offered in the literature, for the purpose of the present work, we adopted three automated methods satisfying the above conditions, namely Climatol, ACMANT and RH Test.

Climatol is a relative homogenization method based on the Standard Normal Homogeneity Test (SNHT) (Alexandersson, 1986), available as an R package (Guijarro, 2023). In Climatol, a breakpoint is detected if SNHT statistics returns a value over a given threshold. Here the default SNHT threshold of 25 is used for air temperature. For precipitation, following Guijarro et al. (2023), we set a lower value of 15, given the higher variability of precipitation and the greater difficulty in detecting inhomogeneities.

ACMANT (Adapted Caussinus-Mestre Algorithm for the homogenization of Networks of climatic Time series: Domonkos, 2015 and Domonkos and Coll, 2017b) is a fully automated method, inheriting the detection process from the PRODIGE method (Caussinus and Mestre, 2004). The number of breaks is estimated with the Caussinus-Lyazrhi criterion (Caussinus and Lyazrhi,





1997), and inhomogeneous periods are corrected using the ANalysis Of VAriance (ANOVA) method. When run in automatic mode, ACMANT requires a set of input parameters and settings concerning outliers filtering, the output format, and the snow season period for precipitation. Here, the snow season is set from November to May, the output files are kept in the default format and the program is run ignoring outliers filtering, because outliers are already removed during the QC process described in the previous sections.

RH Test (Wang, 2008), suggested by the Expert Team on Climate Change Detection and Indices (ETCCDI), detects breakpoints using a penalised maximal T-test and requires a reference time series provided by the user, unlike other methods. Here, reference time series $x_{ref}$ are computed as the weighted mean of candidate references $x$:

$$x_{ref} = \frac{\sum_j w_j x_j}{\sum_j w_j} \tag{5}$$

where $w_j$ are the weights, computed as in eq. (4). Detected breakpoints in a time series have to be all significant. Otherwise, the program should be rerun after removing non-significant breakpoints, and the procedure has to be repeated until all the detected breakpoints are labeled as significant.

The above methods are among those ranked best in comparative studies of different homogenization approaches, such as the Multi-test project (Domonkos and Coll, 2017a; Guijarro et al., 2023). Their use is documented in several studies concerning different climate variables (Luna et al., 2012; Mamara et al., 2013; Azorin-Molina et al., 2016; Chimani et al., 2018; Hunziker et al., 2018; Squintu et al., 2019; Brugnara et al., 2023). Additionally, all three methods have a high tolerance for missing values.

### 3.5 Detection of inhomogeneities

All the above methods employ a relative breakpoint detection approach, i.e. using information from a set of neighbouring stations. Homogeneity methods are run on single time series, monthly aggregated, with at least 30 years of data and 70% of valid observations (Wijngaard et al., 2003). These conditions are also adopted when selecting the set of reference stations for the homogenization process, as reported in table 3. Reference stations are further selected based on horizontal distance, elevation difference, and time correlation, as reported in table 3. Specifically, reference stations must be located within a horizontal radius of 100 km centered on the test station. An elevation difference threshold of 300 m was chosen, a value included within the range of 200-500 m adopted in the literature (e.g. Buchmann et al. (2022)). Reference time series are further selected based on the Pearson's correlation coefficient of first differences, compared to the tested time series. The coefficient is required to be no smaller than 0.9 (Kunert et al., 2024). If no reference station meets this threshold, a time correlation of at least 0.8 is accepted. In case of precipitation, the thresholds are 0.8 and 0.7, respectively. Homogeneity is tested if at least four reference stations can be found. The maximum number of reference stations is set to 25 as an optimal compromise between the reliability of the procedure and a reasonable computational time for the homogenization process. When this threshold is exceeded, only time series with a higher percentage of valid data are retained.

Homogenization results obtained by the three methods are analysed to identify time series requiring corrections as affected by one or more breakpoints (fig. 5). The assessment of breakpoint significance is based on cross-comparison among candidates





identified by more than one method to minimize false positives. Hence, following Buchmann et al. (2022), a breakpoint is
considered significant if at least two methods detect it within the same time window, with a tolerance of $\pm 2$ years. However,
breakpoints in the first and last two years of the series are rejected because all methods typically struggle with interpreting
changes occurring either at the beginning or the end of time series (Ducré-Robitaille et al., 2003; Resch et al., 2023). In
addition, if multiple breakpoints are detected in the same time series within a two-year period, only the most significant is
retained based on SNHT and RH Test results.

In view of determining the minimum number of methods required to identify a break as valid, a sensitivity study is performed.
Two configurations are considered: one with all three methods detecting a given breakpoint (named Exp 1) and another with two
out of three methods detecting it (Exp 2). Note that the Exp 2 configuration closely follows the procedure applied by Brugnara
et al. (2023). The results are then compared with a composite set of homogenized time series provided by MeteoFrance,
MeteoSwiss and Histalp (Auer et al., 2007; Chimani et al., 2023). Exp 1 turned out to be too restrictive, identifying only a low
percentage of inhomogeneous time series (about $10\%$ for precipitation and $26\%$ for temperature). Exp 2 showed an agreement
three times higher than Exp 1, and is therefore adopted here to identify inhomogenous series.

### 3.5.1   Homogenization

Time series affected by significant breakpoints are corrected by applying adjustments to each daily value, calculated from
monthly corrections. Inhomogeneous data are corrected conservatively, i.e., data are adjusted only for periods of evident in-
homogeneity. The method used to correct inhomogeneous time series is the quantile-matching technique proposed by Squintu
et al. (2020). This method applies adjustments of different sizes depending on the magnitude of the value to correct. Thus, as
noted by Brugnara et al. (2023), this approach allows for a more robust correction of extreme records compared to methods
applying the same adjustment to all dates regardless of the recorded intensity. Our approach differs from the original method
suggested by Squintu et al. (2020) in the selection of reference stations and applies only one iteration of the original algorithm,
in agreement with the conservative approach adopted for breakpoint detection. The selection of reference stations follows the
procedure already used for the detection of breakpoints, but here no limitation is set on the maximum number of candidate sta-
tions. However, the method was designed mainly for temperature data. For precipitation, suitable modifications are introduced
(see Appendix B for further details). After making corrections, quality control is carried out by applying the range test to evalu-
ate if adjusted data were still physically consistent. If the corrected values did not pass the test, the correction is rejected, and the
original values are kept. Consistency tests were also applied to temperature data. First, minimum and maximum temperatures
were compared to evaluate if $T_{min} \le T_{max}$ and, if not, they were set as equal. Then, the relationship $T_{min} \le T \le T_{max}$ was
assessed. If mean temperature data did not satisfy this condition and both minimum and maximum temperature time series are
available, $T$ data for the whole time series were computed as the average of $T_{min}$ and $T_{max}$. Otherwise, when only minimum
or maximum temperature data were available, $T$ data that did not pass the test were set equal to $T_{min}$ or $T_{max}$.



## 370  4  Results and Discussion

### 4.1  Quality Control

**Table 4.** Summary of results after QC process for air temperature and precipitation. Each column shows for each data provider the percentage of missing data in the raw time series (MISS), flagged values in time, internal and outliers detection phase of quality control (BASE QC), flagged values during application of spatial tests (SPACE QC), inter-quartile range (IQR) of total flagged values (QC IQR) and valid data at the end of the process (VALID).

| PROVIDER | AIR TEMPERATURE | | | | | PRECIPITATION | | | | |
|---|---|---|---|---|---|---|---|---|---|---|
| | MISS | BASE QC | SPACE QC | QC IQR | VALID | MISS | BASE QC | SPACE QC | QC IQR | VALID |
| ARPA FVG | 1.38 | 0.01 | 0.01 | 0.00 | 98.60 | 1.79 | 0.46 | 0.11 | 0.20 | 97.64 |
| ARPA Lombardia | 12.26 | 0.25 | 0.01 | 0.20 | 87.48 | 9.43 | 3.14 | 0.04 | 0.70 | 87.39 |
| ARPA Piemonte | 0.80 | 0.01 | 0.00 | 0.00 | 99.19 | 3.63 | 0.04 | 0.05 | 0.10 | 96.28 |
| ARPAE | 7.54 | 6.30 | 0.01 | 1.90 | 86.15 | 13.93 | 0.20 | 0.02 | 0.30 | 85.85 |
| ARPAL | 4.06 | 0.03 | 0.00 | 0.00 | 95.91 | 6.13 | 0.17 | 0.06 | 0.20 | 93.64 |
| ARPAV | 1.45 | 0.65 | 0.00 | 0.10 | 97.90 | 1.47 | 0.64 | 0.03 | 0.30 | 97.86 |
| ARSO | 2.63 | 0.60 | 0.00 | 0.35 | 96.77 | 2.55 | 0.98 | 0.04 | 0.10 | 96.43 |
| CHMU | 86.46 | 4.06 | 0.00 | 0.00 | 9.48 | 39.75 | 0.00 | 0.00 | 0.00 | 60.25 |
| DHMZ | 3.42 | 0.00 | 0.00 | 0.00 | 96.58 | 27.33 | 0.06 | 0.00 | 0.00 | 72.61 |
| DWD | 2.76 | 0.01 | 0.00 | 0.00 | 97.23 | 7.93 | 0.01 | 0.01 | 0.00 | 92.05 |
| ECA&D | 4.31 | 0.66 | 0.00 | 0.85 | 95.03 | 0.67 | 65.10 | 0.00 | 61.42 | 34.23 |
| Fondazione Edmund Mach | 6.38 | 0.11 | 0.04 | 0.20 | 93.47 | 4.99 | 0.27 | 0.19 | 0.10 | 94.55 |
| GHCN | 9.35 | 0.35 | 0.00 | 0.05 | 90.30 | 27.35 | 0.00 | 0.00 | 0.00 | 72.65 |
| Meteo AM | 12.99 | 0.85 | 0.00 | 0.48 | 86.16 | 30.01 | 8.55 | 0.00 | 0.03 | 61.44 |
| MeteoFrance | 6.97 | 0.37 | 0.00 | 0.20 | 92.66 | 4.66 | 0.14 | 0.01 | 0.10 | 95.19 |
| MeteoSwiss | 3.45 | 0.05 | 0.01 | 0.00 | 96.49 | 4.69 | 0.11 | 0.04 | 0.10 | 95.16 |
| MeteoTrentino | 8.79 | 0.46 | 0.03 | 0.30 | 90.72 | 8.48 | 0.46 | 0.08 | 0.20 | 90.98 |
| OMSZ | 0.86 | 0.00 | 0.00 | 0.00 | 99.14 | 1.40 | 0.00 | 0.02 | 0.00 | 98.58 |
| Provincia Autonoma di Bolzano | 9.60 | 0.14 | 0.00 | 0.20 | 90.26 | 20.21 | 4.88 | 0.01 | 0.00 | 74.90 |
| Regione Marche | 3.02 | 0.62 | 0.02 | 0.10 | 96.34 | 2.90 | 0.25 | 0.01 | 0.10 | 96.84 |
| Regione Toscana | 3.05 | 0.32 | 0.01 | 0.40 | 96.62 | 2.77 | 0.49 | 0.05 | 0.20 | 96.69 |
| Regione Umbria | 16.80 | 0.15 | 0.01 | 0.10 | 83.04 | | | | | |
| Regione Valle D'Aosta | 1.50 | 0.03 | 0.00 | 0.10 | 98.47 | 17.18 | 0.07 | 0.05 | 0.10 | 82.70 |
| SHMU | 2.05 | 0.01 | 0.00 | 0.00 | 97.94 | 59.77 | 0.35 | 1.13 | 0.40 | 38.75 |
| GeoSphere | 5.01 | 0.00 | 0.00 | 0.00 | 94.99 | 3.96 | 0.00 | 0.01 | 0.00 | 96.03 |
| eHYD | 2.32 | 0.01 | 0.00 | 0.00 | 97.67 | 0.73 | 0.01 | 0.02 | 0.00 | 99.24 |
| EEAR | 8.33 | 0.61 | 0.01 | 0.21 | 91.05 | 12.33 | 3.33 | 0.08 | 2.49 | 84.26 |





The results of the QC procedure for mean air temperature and precipitation are summarized in table 4 by data provider. Results are shown in terms of the percentage of missing values before QC, flagged values after the two steps of QC, interquartile range (IQR) of flagged values, and valid data after QC procedure. On average, the percentage of flagged values is below 1% for mean temperature data. For precipitation, the average value is higher, more than 3%, largely due to the very high percentage of flagged values for ECA&D data, with percentages still exceeding 1% in a few other cases, such as data from Lombardy (ARPA Lombardia) and South Tyrol (Provincia Autonoma di Bolzano). Air temperature time series are affected by non-negligible quality issues, concerning stations mainly located in Emilia-Romagna (ARPAE), for mean temperature, and Lombardy, for minimum and maximum temperatures. Data providers with high percentages of flagged values also exhibit high IQR, suggesting that average statistics for these providers are affected by poor quality of isolated time series.

The final percentage of valid data after QC is, on average, above 90% for mean temperature and 80% for precipitation, indicating an overall good quality for most of the periods included in the dataset. Indeed, 90% and 75% of the total of missing data at the end of the QC process, respectively, for air temperature and precipitation, were already missing in the raw time series. Data providers exhibiting the lowest percentages of valid measurements are primarily located near the domain borders, such as Czech Republic (CHMU) and Umbria (Regione Umbria, Italy). However, a larger number and extended length of precipitation series increase the likelihood of detecting missing or no valid data. This observation is supported by two statistical insights: a) the amount of valid data in precipitation time series is twice as large as that of air temperature, and b) the density of time series with minimal or no missing data is higher for precipitation than for temperature (see Appendix A).

## 4.2 Homogenization

Fig. 6 shows the distribution of breakpoints from 1961 to 2020 for temperature variables and precipitation. Each series cannot include more than one breakpoint per year, allowing us to express the incidence of inhomogeneities as a percentage of the total number of series available each year. The most prominent peak, observed around the early 1990s, is consistent across all temperature variables and reflects the transition from mechanical to automatic weather stations. A smaller peak in the mid-2000s corresponds to the installation of technologically advanced automatic stations. These newer stations were equipped with improved shielding and ventilation systems designed to overcome the issue of temperature overestimation caused by radiation effects (Böhm et al., 2001; Aguilar et al., 2003; Venema et al., 2013). Another notable peak, more pronounced in mean and minimum temperatures, occurs in the early 1980s. In contrast, the distribution of precipitation breakpoints does not clearly indicate periods of measurement changes, likely due to fewer detected breakpoints.

Inhomogeneous time series account for about 20% for air temperature records (18.3%, 21.6% and 20.6% for mean, minimum and maximum temperature respectively), whereas they are fewer for precipitation, i.e. 12% (see Appendix A). The lower incidence in precipitation series may be attributed to the greater difficulty in detecting breakpoints in these time series, which are well known to typically suffer from higher noise levels (Gubler et al., 2017), stemming from spatial and temporal variability in precipitation measurements, as well as from the complexity of accurately measuring precipitation under different environmental conditions (Peterson et al., 1998). Although the overall number of breakpoints detected in air temperature time series is similar, their distribution among data providers, shown in fig. 7, is more varied. Stations located above 2000 m a.s.l. gen-



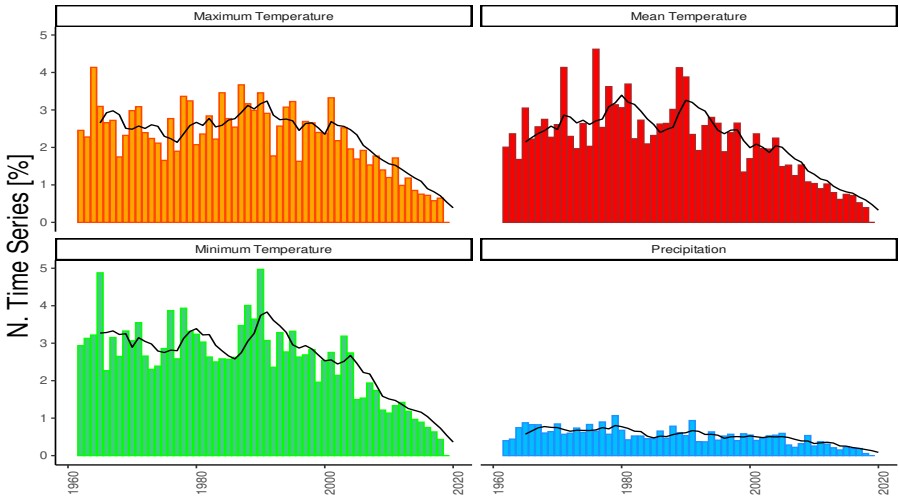

**Figure 6.** Histogram showing the time distribution of detected breakpoints for the period 1961-2020, expressed as a percentage of stations with respect to their total amount. The black line represents the 5-year moving-window average.

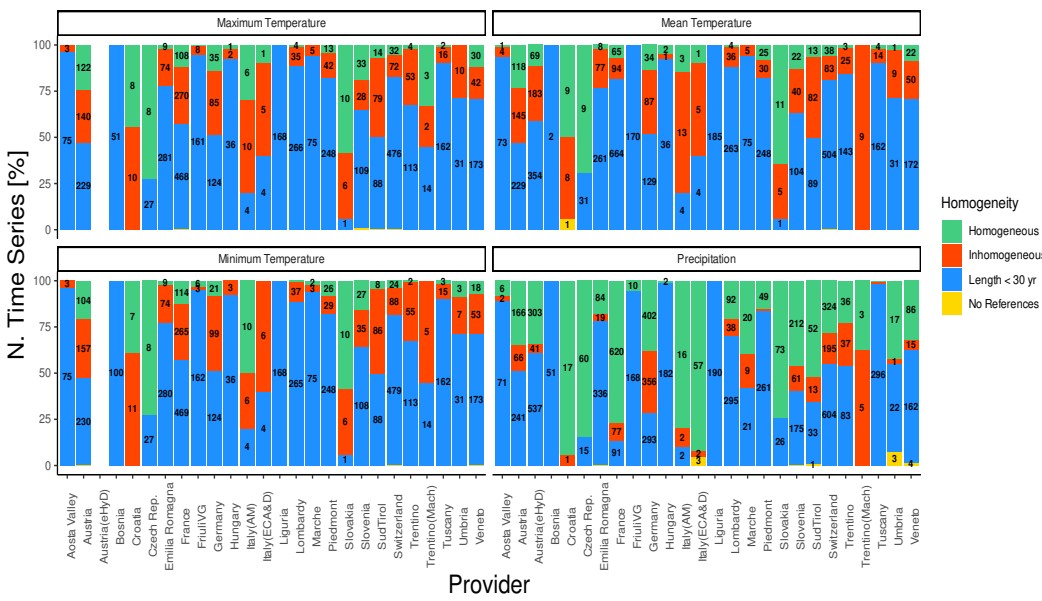

**Figure 7.** Distribution of homogeneous (green), inhomogeneous (red), insufficiently long (blue) and lacking of reference stations (yellow) time series by data provider for mean, minimum and maximum air temperature, and precipitation. The y-axis indicates the percentage of stations in each category, while labels within bars show the absolute amounts.



erally exhibit higher homogeneity (see Appendix A), likely due to fewer time series that are either too short or lack sufficient references for homogeneity testing. The percentage of time series that cannot be tested due to inadequate reference stations is negligible, typically ranging from 0.1 to 0.2%. These situations are primarily associated with very-high elevation sites or regions near domain borders with lower station density.

Fig. 8 shows boxplots of mean daily adjustments applied to each inhomogeneous time series for temperature (mean, maximum and minimum) and precipitation by the data provider. Providers whose time series are not affected by inhomogeneities, or require minimal corrections, are omitted. Average daily adjustments generally center around zero, and fall within $\pm 2°C$ range for air temperature and $\pm 10mm$ for precipitation. However, corrections exceeding these ranges are present, albeit infrequently, as outliers in boxplots for some data providers. France (MeteoFrance), Lombardy (ARPA Lombardia) and Slovenia (ARSO)

show a higher prevalence of time series requiring nonnegligible corrections for both air temperature and precipitation.

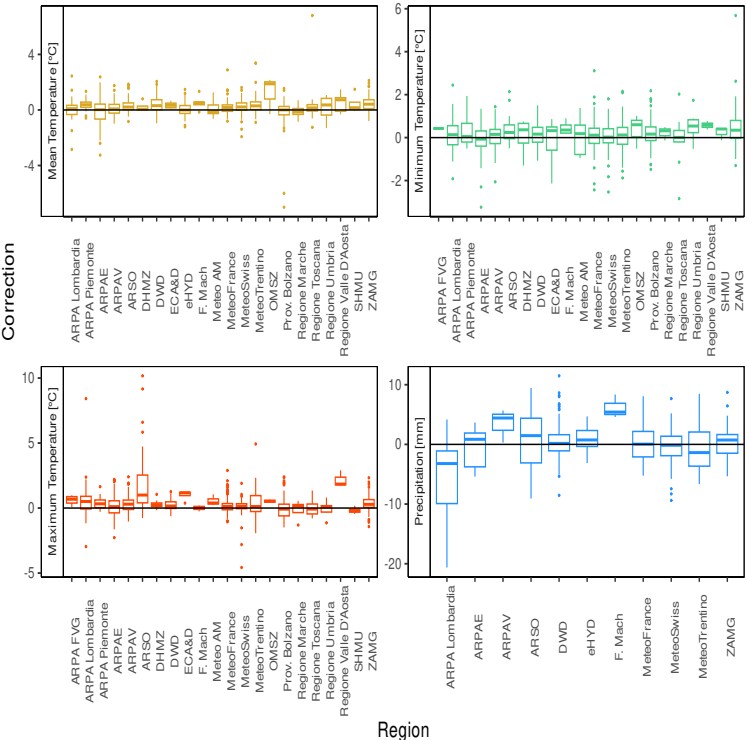

**Figure 8.** Boxplots of mean daily adjustments for each inhomogeneous time series by data provider. Each box includes data within quartiles, and the central line shows the median. Data outside the box, but within 1.5IQR, are represented by upper and lower whiskers. Data exceeding these thresholds, i.e. outliers, are shown as points. A black line indicates the situation when no correction is required.




## 4.3 Dataset Features

The new and unprecedented EEAR-Clim dataset developed in this study features key properties that are different from the state-of-the-art for similar station-based observational products in terms of quality, space-time resolution, and completeness. Quality improvements, attained through the application of an extensive and accurate procedure, and the increased resolution in time and space are expected to provide a more realistic representation of environmental conditions. This has the potential to positively affect the accuracy of hydro-climate predictions and enhance the reliability of bias-corrected model data (Laiti et al., 2018). Moreover, a more realistic understanding of climate variables plays a key role in improving our knowledge of the Alpine climate state and its variability (Hartmann et al., 2013; Begert et al., 2005; Skrynyk et al., 2023). The EEAR-Clim dataset, compared to existing observational products covering the Alpine area, increases the spatial coverage in terms of available time series by more than 30%, allowing for an improved representation of the orographic effects. The higher spatial density also allows for an enhanced representation of climate variability with elevation. Comparing density of stations in different altitudinal ranges, we found similar values converging to an average density of about 1.5 stations every 10 $km^2$, implying a rather homogeneous distribution of observations across elevation ranges. A reliable comparison in terms of elevation distribution with other products cannot be easily achieved: the strong decay of available observations at higher elevations is widely known (de Jong, 2015). The multi-parameter feature of our dataset is rare to find in high-resolution observational products. Thus, the collection of high amount of data for several meteorological variables at daily resolution is really unprecedented and it can enhance an integrated assessment of Alpine climate changes based on a better understanding of interactions between the different variables (Brunetti et al., 2009; Gaffen and Ross, 1999; Kaiser, 2000; Wang and Gaffen, 2001; Huth and Pokorná, 2005; Beniston, 2006). Our collection efforts also resulted in gathering observations with enough time coverage. Most of them consist of at least 60 years of data and a part of observations extend up to a century, despite many historical records are unavailable in digital form yet. Different initiatives of historical data digitization are underway, thus the dataset can be further expanded in terms of data coverage beyond the last 60 years.

## 5 Conclusions

A new observational dataset of air temperature and precipitation at daily resolution for the Extended European Alpine Region (EEAR), covering the whole available period of measurements, has been presented. The data collection effort resulted in a very high spatial density and led to a homogeneous regional coverage. This achievement was favoured by newly digitized data and the collection of datasets from national, regional and local institutions. The EEAR-Clim dataset includes most of the daily available stations in the EEAR, managing to increase the density even at higher elevations, which is a typical issue of observational datasets in topographically complex regions. Furthermore, collecting data from multi-variable measurements and including the most recent records (updated to the year 2020) are important add-on improvements compared to other available products for the area. Here, the dataset consists of air temperature and precipitation data, but an updated version also including additional variables is planned to be released. Substantial efforts were made to ensure the consistency and quality of the different data contributions. A deep and extensive quality control was carried out, following WMO criteria in

terms of data quality (WMO, 2017), merging different approaches and integrating new techniques aimed at facing all critical
issues of rescued data. The QC procedure flagged about $5\%$ of total observations, of which $80\%$ are precipitation data. While
QC did not significantly affect the amount of valid data, that is about $90\%$ on average, it improved the overall accuracy. A
tailored homogenization procedure was performed on quality checked data. The break detection stage was based on three
automatic methods: Climatol, ACMANT and RH Test. The comparison of results provided by these independent procedures
ensures reliable identification of significant change-points, especially when metadata are not available (Fioravanti et al., 2019).
Inhomogeneous time series with identified breakpoints were homogenized using the quantile-matching algorithm, applying
adjustments depending on percentiles of the empirical distribution. The inhomogeneities detected in precipitation time series
are fewer than those identified in temperature time series, as also reported in other studies (Gubler et al., 2017; Skrynyk et al.,
2023). The increased homogeneity at elevations above 2000 m a.s.l. could be explained by external factors (e.g. a reduced
sample of stations) rather than specific accuracy of high-elevation time series. Breakpoints detection results and adjustments
magnitudes were in agreement with other existing studies focused on areas including the EEAR or its sub-portions (Brugnara
et al., 2023; Squintu et al., 2020; Mamara et al., 2013; Coll et al., 2020). A subset of the time series covering the 1961-
2020 period was used as a basis to carry out an extended analysis of trends and climate features of both average values
and extremes. Additionally, a high-resolution interpolated version of the EEAR-Clim dataset is planned for release. These
subsequent analyses and applications highlight the relevance of the new observational dataset developed in this work as a
tool for better understanding Alpine climate changes over recent decades and improving the reliability of model simulations
and future scenarios. The procedure developed within this work can be readily implemented over other areas or time periods,
adapted to time series at different time frequencies and extended to other variables, such as relative humidity, wind speed, solar
radiation or snow depth.

## 6    Code and data availability

All computations were performed with R statistical software version 4.2.1 (R Core Team, 2022). The code is available from a
repository, including the main scripts to read and process data, perform quality control and the main tasks of homogenization.
Most of the contributing institutions agreed to share their data (see table 1). Hence the open data are available from Zenodo
repository (https://doi.org/10.5281/zenodo.10951610, Bongiovanni et al., 2024) as raw, quality checked and homogenized time
series. For the full dataset, including undisclosed data, please contact the corresponding author.

**Appendix A:  Additional Material**

**Appendix B:  Correction of Inhomogeneous Precipitation Data**

The calculation of adjustments for precipitation time series affected by identified inhomogeneities follows the quantile-based
procedure suggested by Squintu et al. (2019), but adapted for precipitation data. Values below 0.1 mm are not corrected. The
computation of the adjustment factor $a_{i,j,q,m}$ (eq. 2 in Squintu et al. (2019)) and the final adjusted value $\bar{v}$ (eq. 4 and 5 Squintu

**Table A1.** Summary table of break detection results based on the multi-methods comparison. Amount and related percentages of homogeneous, inhomogeneous and not tested time series are reported. Not tested time series are grouped in those with an extent below 30 years and without enough reference stations.

|  | **T** | **Tmin** | **Tmax** | **TP** |
|---|---|---|---|---|
| *Homogeneous* | 462 (8.4%) | 404 (8.4%) | 444 (9.2%) | 2710 (34.5%) |
| *Inhomogeneous* | 1005 (18.3%) | 1046 (21.6%) | 997 (20.6%) | 945 (12.0%) |
| *Length<30yr* | 4016 (73.2%) | 3392 (69.9%) | 3391 (70.1%) | 4201 (53.3%) |
| *No References* | 3 (0.1%) | 3 (0.1%) | 6 (0.1%) | 13 (0.2%) |
| Total | *5486 (100%)* | *4845 (100%)* | *4838 (100%)* | *7869 (100%)* |

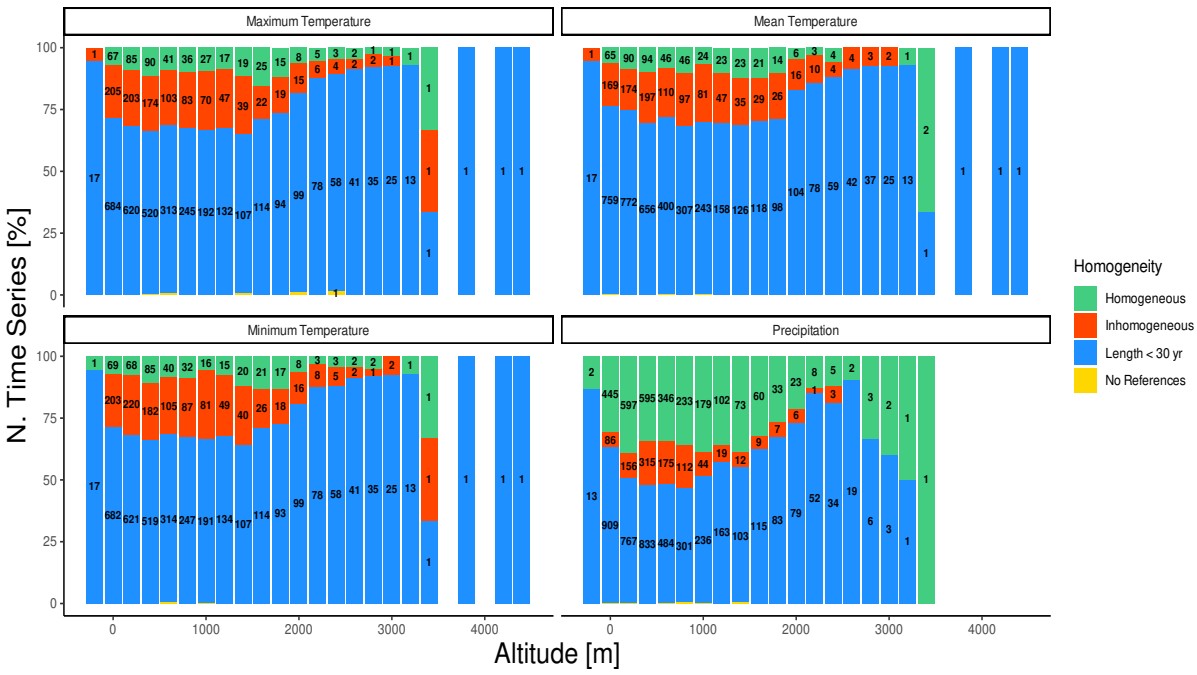

**Figure A1.** Distribution of homogeneous (green), inhomogeneous (red), insufficiently long (blue) and without references (yellow) time series over elevation for mean, minimum and maximum air temperature, and precipitation. The y-axis indicates the percentage of stations in each category, while numbers within bars report the absolute amounts.





et al. (2019)) have been modified accordingly. In particular, the adjustment factor $a_{i,j,q,m}$ is computed as follow:

$$a_{i,j,q,m} = \frac{\frac{b_{q,m}}{r_{j,q,m}^{aft}}}{\frac{s_{i,q,m}}{r_{j,q,m}^{bef}}} \tag{B1}$$

Instead, the final adjustments are computed as the median over $j$ values:

$$\tilde{v}_j = v * a_{j,\tilde{q},m} \tag{B2}$$

thus simply converting the temperature formula from additive to multiplicative, making it suitable for precipitation data.

*Author contributions.* The original idea of the work was conceived by GB, DZ and BM. Quality control and homogenization procedures
were performed by GB with the help of AC and MM. The analysis of results was performed by GB. The first draft of the paper was prepared
by GB. All co-authors revised and refined the manuscript.

*Competing interests.* The contact author has declared that none of the authors has any competing interests.

*Acknowledgements.* This paper and the related research have been conducted during and with the support of the Italian national inter-
490 university Doctoral Programme in Sustainable Development and Climate change (www.phd-sdc.it), supported by a grant co-funded by the
Programme and by the University of Trento. AN has been supported by the Fondazione CARITRO, Bando Post-DOC 2022. We acknowledge
all data providers for kindly providing the data. In particular the authors thank all the personnel who facilitated the data collection, especially
Andrea Pascucci (Regione Umbria), Luca Maraldo (Provincia Autonoma di Bolzano), Denise Ponziani (Regione Valle d'Aosta), Kristina
Szaboova (Slovenský hydrometeorologický ústav), Eva Mandl (Hungarian Meteorological Service), Maria Bassi (ARPA Piemonte), Damir
Mlinek (Croatian meteorological and hydrological service), Alessandro Biasi (Fondazione Edmund Mach), Luca Rusca (ARPA Liguria),
Raffaele Bertin (ARPA Veneto), Paolo del Santo (Regione Toscana), Marco Pellegrini (Regione Marche).





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
