# Peer review of "EEAR-Clim: A high density observational dataset of daily precipitation and air temperature for the Extended European Alpine Region"

_Earth System Science Data, 2024_

## Author Response (AR1)

Dear Editor,

Thank you for offering us the opportunity to submit a revised version of the manuscript. Please find enclosed the revised manuscript "EEAR-Clim: A high-density observational dataset of daily precipitation and air temperature for the Extended European Alpine Region"), to Earth System Science Data.

The manuscript has been revised addressing all comments and suggestions raised by the reviewers. Please find below our point-by-point replies (in italics) to the Reviewers' comments (in bold), with line numbers referring to the track change version of the manuscript we provided.

The main concerns of reviewers were a better clarification of the novelty of the dataset as compared to other available products, and the provision of guidelines for potential future users. This latter aspect was in particular dealt with by adding the new section "Dataset features and user guide", which replaces the previous "Dataset features". Given the different nature of datasets suggested by reviewers for comparison, gridded or historical-oriented, a quantitative comparison was impossible. We thus focused on highlighting the strengths and weaknesses of EEAR-Clim trying to provide a guide to potential users with a clearer picture of the added value of our dataset. As suggested by reviewer #1, we added details about the possibility of operationalising our dataset.

We refined some aspects, raised by reviewer #2, in the procedures adopted that were not clear enough, especially for final users, adding the required clarifications in the main text. All the other minor suggestions/comments of the reviewers have been addressed.

We hope that the revised version of the manuscript will be positively considered.

With our warmest regards,

Giulio Bongiovanni on behalf of all co-authors.

**RESPONSE TO RC1**

*The authors thank the reviewer for thoroughly reviewing the manuscript. We are also grateful for acknowledging the recognition of the value of the EEAR-Clim dataset for both the scientific community and climate services providers. Please find our point-by-point replies (in italics) to your comments below (in bold).*

**General Comments:**

**Perhaps it is relevant to add that the publication of this manuscript and the dataset sparked interest at the climatological departments of MeteoSwiss and GeoSphere. And it is in this direction that I have my most important concern. While the authors rightly note in the introduction that the EEAR fills a gap in the dataset landscape that urgently needs to be filled, the EEAR is not completely independent from the existing datasets. Specifically, the reader who is active in Alpine climate research will be interested in the differences between EEAR and existing datasets, like the APGD for a pan-Alpine view on precipitation. Also, GeoSphere published quality controlled and**

**homogenized series for Austria using a somewhat different approach than done in this study. What is missing in the study is an initial assessment of differences between the key existing datasets and the EEAR. Clearly, it will not be possible to make a comparison against all national datasets, but the team of authors is well acquainted with the existing dataset landscape to select a few (like 2) datasets to make a meaningful initial assessment of differences.**

*Reply: EEAR-Clim is definitely related to other national datasets since we collected the station records from the national/regional archives of EEAR countries. The only aspect we could directly compare is the post-processed series after quality checks and homogenization, however, such a comparison is beyond the scope of this study, given ongoing projects focusing on these aspects (e.g. see Guijarro et al., 2023, https://www.climatol.eu/MULTITEST/). Regarding gridded products like APGD, direct comparisons are also challenging, as EEAR-Clim is a station database and not gridded. However, a gridded version of EEAR-Clim is planned to be released, and for that, a comprehensive comparison with existing products will then be possible. However, in line with the reviewer's comment, we decided to better emphasise the strengths and weaknesses of EEAR-Clim relative to existing products: this is now present in the section "Dataset features and users guide" section (L. 456-486).*

**The second concern is actually more of an invitation to contemplate the notion of making this an operational dataset and making it a living dataset rather than a dataset which is very good but is not updated. I would like to urge the authors to add a section in the manuscript discussing the possibility and requirements to make this dataset operational. The guess of this reviewer is that one bottleneck in making the dataset operational is access to data. Perhaps the authors may want to refer to recent initiatives, like the EUMETNET Rodeo project https://rodeo-project.eu/ which build upon the High-Value Datasets directive of the European Commission. (https://digital-strategy.ec.europa.eu/en/news/commission-defines-high-value-datasets -be-made-available-re-use)**

*Reply: The EEAR-Clim dataset was developed within a doctoral program supported by academic institutions, namely IUSS and the University of Trento, which are not service agencies and cannot face this kind of initiative. While we recognize the immense added value of making this dataset into an operational, regularly updated dataset, such an endeavour requires permanent and funded projects, open data access frameworks, and supportive European policies. We acknowledge that ongoing initiatives, such as those inspired by the European Commission's High-Value Datasets directive, and projects like the EUMETNET ROdeo represent a promising direction. However, at the moment, our plan is to update the dataset in the future, including time series extending back in the past as well as measurements of other variables. In the manuscript, we added further information about what is needed for making EEAR-Clim operational, emphasizing current limitations, and clarified our plans for the dataset update (L497-502).*

**Other issues the authors may want to look into**

**\* line 32. The Hofstra et al. and Kysely & Plavcova publications refer to rather ancient versions of E-OBS. It would be appreciated if a remark could be added which makes clear that station density in this part of the greater Alpine region (and in other parts) has improved significantly. Although it remains fair to state that EEAR still has a higher stations density in many parts of the greater Alpine area.**

*Reply: We agree that Hofstra et al. and Kysely & Plavcova publications refer to versions of E-OBS with a lower density compared to the current state of the dataset presented in Cornes et al (2018) ([https://doi.org/10.1029/2017JD028200](https://doi.org/10.1029/2017JD028200)) in which many more station series have been added with an increase from 1,200 to approximately 3,700 stations for temperature, and from 2,500 to approximately 9000 stations in the case of precipitation over Europe. However, this significant increase is not homogeneous, including parts of the Alpine range, and it does not solve the disparity in density between northern and southern Alps. The Italian Alps, but not only, is the most critical area, where the difference in density of weather stations is more marked, as you fairly said. We have added a few lines in the main text to acknowledge your remark and highlight the improvement of the E-OBS dataset (L.36-37).*

**\* lines 214-221. The issue of time shift is indeed an important issue. The 3-day moving window approach alleviates this to some degree, for example if a 24-h accumulated rainfall amount is attached to the date of the start of the 24-h period where the reference has attached this amount to the end date of the period. However, it is likely that the reference and target station provide data that partially overlap. For these cases it is less clear if the sliding window approach works. In this case, sometimes accumulated precip of the previous (or next) day will match better than the amounts recorded on days with the same data stamp. Would it be possible for the authors to spend a few more words on this subject and how this has been approached?**

**Perhaps it is worthwhile to check https://doi.org/10.5194/essd-15-1441-2023 where a similar problem is encountered.**

*Reply: The 3-day moving window approach is designed to address this issue effectively. Specifically, each measurement in the target station is compared to the values recorded at neighbouring stations on the previous, current and subsequent day. This ensures that the procedure accounts for potential discrepancies in how 24-hour accumulated rainfall is assigned (e.g., whether it is attached to the start or end of the accumulation period). This method is especially effective in "Wet and Dry Isolated Reports Test" (section 3.3.1, L250-269), where this approach helps to compare wetness conditions. We have clarified this aspect further in the manuscript, adding a more explicit description of how the 3-day moving window addresses the time-shifting issue (L. 239-241).*

**very minor issues**

**\* line 140: typo in 'metrics' (is plural, should be single)**

*Reply: The typo has been corrected in the manuscript.*

* table 2: the description of the outliers detection in the table and in the text (line 210-212) could be more complete. The table adds criteria on temperature which are not described in the text.

*Reply:* *We acknowledge that the criteria for temperature, especially in the case of the outlier detection formula, were not accurately described in the text due to an oversight. We have corrected this and revised the text to align with the formula presented in Table 2.*

* line 311: Sadly, the ETCCDI does no longer exist as a WMO Expert Team. It has been replaced by ET-SCI (Sector specific Climate Indices) https://climpact-sci.org/assets/etsci/etsci-poster-20190220-en.pdf Not much activity seems to come from this Expert Team. All I can suggest is to add a link to the https://etccdi.pacificclimate.org/ website to guide the unaware reader.

*Reply:* *We have added the suggested link to the manuscript to guide readers.*

* line 489: It would be appreciated if data providers to ECA&D were acknowledged here as well (https://knmi-ecad-assets-prd.s3.amazonaws.com/documents/ECAD_datapolicy.pdf)

*Reply:* *We have added an acknowledgement for the data providers of the ECA&D in accordance with their data policy.*

**RESPONSE TO RC2**

*The authors thank the reviewer for the thorough review of the manuscript. We are grateful for your positive evaluation of our work and your recognition of the value of the EEAR-Clim dataset. We also appreciate your detailed insights as an end user of meteorological data and for your helpful suggestions on improving the clarity and depth of our dataset description. Please find our point-by-point answers to your comments below.*

**General comments**

1. The dataset is large (9000 series), with a complicated time structure (essentially series shorter than 30 years, but also long ones; their overlap is not clear). The quality check and corrections are extensive, but homogenisation was applied to less than 20% of the series (12% for P). Hence the dataset is still quite heterogeneous, and i feel that more explanations, or even advice, could be given to potential users, especially strengths and weaknesses compared to other datasets. For instance, it would be interesting to know which part of the dataset is considered more accurate (e.g., when an absolute threshold like 0ºC is considered), and which part is considered less accurate (but still useful for statistical analysis).

*Reply: We have recognised that the statement in the original manuscript about homogenization coverage was not as clear as it could have been. Homogenization was applied to about 30% temperature and 46% precipitation series covering all time series longer than 30 years. Among these, 20% for T and 12% for P were found inhomogeneous and corrected accordingly. This procedure guarantees that all time series longer than 30 years can be used to study climate trends or variability with enhanced reliability. Thus, this is not an index of heterogeneity of the dataset, but rather of homogeneity and accuracy. About time series shorter than 30 years, in that case, we followed the same approach of other homogenization studies. Indeed, in the case of shorter time series break detection methods provide less reliable results, with a lot of uncertainty whether identified breakpoints are reliable or only algorithm artifacts. Thus, the exclusion of time series from homogenization prevents inclusion of more uncertainties in the dataset and increases data confidence. We added further explanations in the manuscript, clarifying better the implications of the approach adopted, and the most suitable use in each case (L 474-490).*

**Two specific comments:**

**1.1 Authors underline the possibility to study climatic change from their dataset but 70% of the T series, and more than half of the P series, are shorter than 30 years, the duration required to calculate a climatic mean (WMO definition). Hence to me the largest part of the dataset cannot be used to study climatic trends, but rather climatic interannual to decadal variability. This should be underlined.**

*Reply: While it is true that part of the series does not reach the 30-year threshold to define a climatic mean in agreement to the WMO standards, the dataset still includes approximately 1600 temperature and 4000 precipitation series satisfying this requirement. This represents a substantial improvement compared to station-based datasets used in previous studies such as HISTALP (Auer et al. 2005) consisting of about 242 long-term series, or other national products with similar amounts of stations. This information is now reported in the revised version of the manuscript (L. 480-490).*

**1.2 A homogenisation procedure has been applied, and its effect on long term trends is not clear to me: if the correction were perfect, the procedure should preserve these trends. However, in practice, the correction of the 'non-climatic' change is not perfect, and this may increase or decrease the long term trend. I guess this effect has been tested in previous studies, by applying other procedures (use of metadata; least-square fitting with nearby stations, etc.). Maybe some indications could be given.**

*Reply: As the reviewer remarked, homogenization is a critical step to ensure the reliability of long-term trends. Homogenization aims to preserve the real climatic signal, not affected by artificial inhomogeneities caused by non-climatic factors such as changes in instrumentation, station relocation, or observer practices. As already observed by several studies (e.g., Begert et al., 2005; Curci et al., 2021; Brugnara et al., 2022; Guijarro, 2023), in inhomogeneous time series the original signal tends to exhibit unrealistic oscillations, underestimating and reducing the spatial coherence of trends, as well as showing higher variability in anomalies. The methods we employed were selected based on their performance and ability to distinguish artificial inhomogeneities from natural variability effectively. While no*

*homogenization procedure is perfect, the adjustments generally lead to more accurate representations of the real climatic evolution, reducing biases in trend estimates. We have added further explanations about this aspect in the manuscript (L 377-380).*

**2. The whole interest of the dataset is to compare series in different area. However, techniques and procedures of measurements have been widely different between operators, with strong differences due to both the daily resolution and the complex mountain environment. How far can the series of this dataset be compared; can some systematic differences be expected? I think this point should be addressed, since most users of the dataset will have no clue about this question.**

*Reply: Systematic differences are indeed present given the varying measurement techniques and procedures across operators, and the challenges posed by the complex mountain environment. For example, the computation of mean temperature is not uniform among data providers: some calculate it as the mean of minimum and maximum temperature, others as the average of values measured at hourly intervals or other sub-daily time scales. For mean temperature, we reduced these discrepancies by computing the daily average from hourly values, when possible depending on their availability. An enhanced comparability among temperature time series also resulted after the application of quality control and homogenization procedures, including the post-homogenization control to ensure the reliability of the corrected time series (see L for further details). We added this information in the main text (L490-496).*

**At least since the 1990s the measurement techniques and procedures should be well known. I guess measurement techniques did converge over the recent decades, and this concerns most of the data, but did the measurement procedures also converge (e.g. times of measurement in the day)? Some technical knowledge was also acquired with intercomparison projects (esp. the ones lead by WMO). This problem is especially sensitive for precipitation and extremes; it is also sensitive because the dataset aggregates mountain and plain area, where techniques and measurement biases are different.**

*Reply: As the reviewer mentioned, precipitation is one of the variables that is strongly sensitive to different kinds of techniques and measurements, for example, due to the need to use heated rain gauges at higher elevations and the differences in temporal aggregation intervals. Metadata should help to identify these discrepancies, but this is impossible in practice due to the limited access to them and the lack of important information (e.g. heated rain gauge information rarely reported). Despite intercomparison initiatives, such as the ones led by WMO, there is not an agreement among data providers yet. The QC procedure implemented in this work aims to strongly reduce anomalous precipitation measurements for stations not equipped with heated rain gauges. We revised the manuscript (L490-496) to include this information and provide guidance for dataset users.*

**2.1 Difference in measurement technique is especially a problem for precipitation in mountain area, due to a potentially strong wind bias, and even more for estimating snow water equivalent, which can be very tricky. WMO has organised intercomparison**

**projects, since at least the 1980s (Sevruk et al.): did it help to homogenise the measurement techniques? Since then?**

*Reply: As already stated in the previous comment (2.), the lack of available information about equipment with heated rain gauges in metadata makes the identification of these discrepancies very difficult. Intercomparison projects (Sevruk et al.), partially helped to homogenize measuring techniques after the 1980s, but not-negligible discrepancies are still present. Whether most of the stations now should be equipped with heated rain gauges, the time at which precipitation measurements are assigned is still heterogeneous. The study of homogeneity in precipitation time series also revealed no clear difference in the number of inhomogeneities before and after the 1980s, highlighting that probably the main issues in precipitation recording are not properly addressed. A reversed tendency, and a possible enhancement in this sense, have been noted in the last 5-10 years, but it requires further assessments in the future.*

**2.2 Measurements procedures have been also widely different. The case of precipitation is the most sensitive since a precipitation event can be spread over few hours to few days: the measurement procedure could be to read the daily total at 6am, and the value attributed to the previous day, or to read the total at 6pm and the value attributed to the same day, so that a precipitation event could be cut and spread over different days. With a strong impact on precipitation intensity.**

*Reply: We addressed the time shifting in precipitation observations by applying a 3-day moving window approach, specifically designed to overcome this issue. In detail, each measurement is compared to the values recorded at neighbouring stations on the previous, current and next day. This ensures to consider potential discrepancies in the day at which 24-hour accumulated rainfall is assigned (e.g., whether it is attached to the start or end of the accumulation period). This method is especially effective when we compare wetness conditions in tested and reference stations, for example, during quality control. We have better clarified this aspect further in the revised manuscript (L239-241).*

**3. The structure of the dataset is not yet clear to me.**

**3.1 Time structure. I do not understand what is actually displayed by Figure 2a. I expected Fig.2a-b to be cumulative distributions (esp. because continuous lines/curves suggest a time continuity). Obviously this is not the case, but Fig2a is not clear to me, even with the indication (L.132) that these are "10-yr increments". The first value of stations number seems to correspond to a record length of 1 year, is that the number of stations with a record length of 1 to 10 years? More explanations should be given in the figure legend, esp. by given an example for both figures. I guess Fig2a would be clearer with vertical bars.**

*Reply: Both figures are indeed cumulative distributions. Figure 2a shows the distribution of the number of stations as a function of the minimum length of their time series, expressed through 10-year increments, considering all available data from 1870 to 2020. For example, at x=30 years, the figure shows the number of time series covering at least 30 years (not necessarily within the 1991–2020 period). Similarly, at x=40 years, the figure shows the*

*number of stations with time series covering at least 40 years, and so on. While we considered plotting Figure 2a as vertical bars, merging results for all variables in this format made the figure more cluttered. To address your concern, we improved the main text (L 139-141), and we added a better description of the figure in the caption to clarify and explain the content of Figure 2a.*

**3.2 Further, comparing both figures 2a and b, i have the impression that the longest series are not necessarily the latest. Hence the information of the series overlap is missing in these figures, and it would be great to see some information on their overlap (maybe with an additional figure).**

*Reply: The reviewer is right, the longest series are not necessarily the latest, and, in addition, shorter series are not exclusively from 1991 to 2020. This reflects the complexity of the dataset structure, which results in a variable overlap of time series. For clarity, a heatmap showing how the stations are distributed in time is shown here: the plot below represents the overlap of the dataset time series, with the time extent highlighted by red bars. Recent time periods are characterised by shorter time series compared to earlier periods. However, in a few cases, stations worked for a limited period (e.g. see the group of stations around 1900) and then were dismissed or replaced. In other cases, it can be noticed that longer time series, covering 50-60 years, started around 1950-1960s and stopped in the 2000s. Thus, each time series has an independent time extent leading to a very complex time structure of the dataset, even more complex due to the high number of stations. The plot confirms the dataset complexity and variety. With this kind of dataset, it is not possible to establish a complete structure of the dataset, but it is certainly possible to provide additional information about it: in L 148-150, we have added useful information about the overlap.*

[Figure]

*Heatmap showing the complex time structure of EEAR-Clim. The time extent of each station is highlighted by red colors, limited to the period 1840-2020.*

**3.3 Spatial structure. How the area has been defined? The whole focus of the study is about "mountain terrain", but Fig.3b shows that, in fact, most of the area has an elevation lower than, say, 400m, and most of the stations are below 500m (L.149). This seems at odd with the claim that (L.108) "EEAR is predominantly constituted by**

complex terrain and hence characterized by strong elevation gradients". I suggest to give some clue about the area definition (why 3-18E, 43-49N ?), and to qualify the above comments.

*Reply: The area and its extent (3-18°E, 43-49°N) define the Extended European Alpine Region (EEAR), with the word "extended" meaning that it is not limited to the European Alps, but it includes also surrounding regions, such as the pre-Alpine areas and nearby plains, which are climatically linked to the Alpine system. This extent is very similar to the Greater Alpine Region (GAR), widely used in Alpine climate studies, but shifted 1° westward to better include the French Prealps, which we consider more relevant to Alpine climate studies than the Hungarian plain. Regarding the comment about the complexity of the terrain, we have to highlight that "complex terrain" does not exclusively refer to mountainous areas but in general for regions with irregular orography (https://glossarytest.ametsoc.net/wiki/Complex_terrain). For example, coastlines are recognized as complex terrain areas. The EEAR encompasses mountains, plains, hills, coastlines, and valleys of different depths and widths, making it one of Europe's most orographically complex regions. This complexity is reflected in the elevation range, from -5 m a.s.l. to 4807 m a.s.l., as already stated in the manuscript. Focusing on elevation distribution, most of the area has an elevation below 800 m (approximately), due to the presence of large plains included between mountain ranges (e.g. Po Valley between the Italian Alps and Apennines). This does not contrast with the concept of complex nature, but supports it: the complexity of the regional orography is enhanced by the presence of large plains and high mountain massifs. Concerning station distribution, we clarified that more than 50% of stations are located below 500 m a.s.l., whilst a relatively high percentage of 40% of the stations are located between 500 and 1500 metres. We better clarified this in the manuscript to avoid misinterpretations (see line 162-164). Although 50% of stations are below 500 m and only 10% above 1500 m, the average density per $km^2$ is similar because, as said before, the lowest elevations cover a larger area. Moreover, the density of stations per area in the lowest and highest elevations is similar, as stated in the main text (L164-165).*

**B. Specific comments**

**Abstract:**

**L.7-8 The fact that most of the 9000 series are short (less than 30 years) and restricted to the period 1991-2020 should be underlined.**

*Reply: Shorter time series are not exclusively restricted to 1991–2020, as you noted in comment 3.2. For example, some weather stations ceased operation in the 1980s or earlier due to e.g., site replacements, technological upgrades, or instrumentation damage. However, we revised the text to underscore that 1991-2020 is the most covered period by measurements (L 9).*

**L.14-15: "better understanding climate change" and climatic variability, mostly, since most of the series are too short to address a climatic trend (i.e., over less than 30 years)**

*Reply: We have added "climate variability" in this sentence (L. 20).*

**L.16 "The continuous warming of the climate" > "Global warming" [suggestion]**

*Reply: We have accepted your suggestion and corrected it in the manuscript.*

**L.21-22 "benefits from a density of weather stations and length of data series not easily attainable in many other regions": not clear neither complete to me. I guess what is meant is the availability of many stations and long duration of series. But this richness should be compared to the heterogeneity of the mountain environment, which requires much more stations to resolve "the complex nature of Alpine terrain" (L.37).**

*Reply: Here we only refer to mountain regions, and the observational network of the Alps is among the best-developed compared to other important mountain ranges (Himalaya, Andes etc.). We agree that to resolve the complex nature of Alpine terrain, we need many more stations, and in the last decades, the scientific community has moved forward in this direction. However, in the meantime, we should use data from all available stations, understanding strengths and weaknesses, because it is useless to have many more stations without properly using their data. In any case, we modified the sentence (L. 25-26) to clarify we are referring to a comparison of the weather network of the Alps with that of other mountain regions.*

**L.31 E-OBS is a gridded product, it would be interesting to have some indications of strengths/weaknesses compared to this dataset.**

*Reply: Thank you for your suggestion. E-OBS and EEAR-Clim are different kinds of products as E-OBS is gridded, while the latter is station-based, which leads to distinct strengths and weaknesses depending on the intended application. The spatial resolution, covered period, considered area, and the type of analysis or application define the suitability of the dataset. EEAR-Clim compared to E-OBS share similar features, such as a similar homogenization procedure, period covered, and daily data resolution. However, EEAR-Clim strongly improves the spatial density, especially in areas where the station database beyond E-OBS is lacking such as the Italian Alps or high-elevated areas, includes a subset of time series extending beyond the 1950 in the past, and benefits of a more detailed quality control. A detailed discussion about the main strengths and weaknesses of EEAR-Clim in comparison with other products is now provided in the "Dataset Features and User Guide" section, including a qualitative analysis. Further information is also included in the introduction section of the manuscript.*

**L.32 a word missing here, i guess "has a lower density"**

*Reply: We corrected the mistake in the main text.*

**L.53 "intensity of extreme weather events" is very sensitive to the measurement procedure (cf. Comment 1.2), hence the need to document the procedure as metadata.**

*Reply: As mentioned in comments 2-2.1, the limited access to metadata makes it difficult to document systematic differences in measurement procedures. The homogenization procedure we followed should strongly reduce this sensitivity, enhancing the confidence in data time series (L. 476-479).*

**L.58-60 "data quality", "accurate data", is a bit too vague to me, quality depends on the use of data.**

*Reply: We understand your concern. In our opinion data quality plays a key role in any use of the data. Indeed, without exception, low-quality data can compromise the validity of results across all applications. To clarify this concept, we have revised the text in the manuscript (L64).*

**L.125 "TP" for total precipitation is a bit confusing to me, since T refers to temperature; why not P instead?**

*Reply: We replaced "TP" with "P" throughout the manuscript to avoid confusion with temperature.*

**L.125 "global provider" (GHCN): maybe make it clear that it is only a provider, not a producer, of meteorological observations**

*Reply: To avoid any misunderstanding, we have ensured consistency in terminology throughout the manuscript.*

**L.126 Table 1: ECA&D is a provider, not a producer of data. Why was it used to get data? I guess these series were already quality checked and homogenised, if so, were the procedures similar to the ones used in this work?**

*Reply: As noted, we have referred to ECA&D as a data provider, not a producer, throughout the manuscript. ECA&D was used as a data source because it provides specific time series that were not accessible from other providers. However, as shown in Table 1, the number of stations sourced from ECA&D is relatively small compared to other providers. It is also important to note that ECA&D data are not quality-checked or homogenized. This is evident from Table 4 and Figure 7, which summarise the quality control and homogenization results grouped by data provider.*

**L.126 Table 1 & Status of data: further explanations should be given about the data availability, rather than just the 'open data' flag. What means "available without restrictions"? If about 2700 series are not included in the dataset, then why considering them? (Hopefully restrictions concern raw data and not corrected/homogenised data?)**

*Reply: While recent updates have allowed more time series to be made available as open data, some time series cannot be shared due to data providers' policies. Restrictions apply to raw, quality-checked, and homogenized time series. However, we supplied metadata for these stations alongside all scripts necessary for quality control and homogenization. This approach allows users to process the restricted time series themselves once they collect the*

*raw data from the data provider. If changes to data availability occur, we are committed to updating the dataset accordingly and releasing additional time series as open data where possible. We have clarified the meaning of "open data" in the table caption. Additionally, detailed information about data availability and licensing is provided in the "license.pdf" file in the open repository.*

**L.140 "distance between stations is a useful metrics" not really in mountain area, but at least the simplest metrics**

*Reply: We agree with this consideration. However, our intent was only to give an indication of the densest areas using the distance metric. We clarified this purpose in the text (L 152-153).*

**L.166 daily averages: please make it clear over which period sub-daily data have been averaged (0-24h, 6h-6h, other?)**

*Reply: The period over which sub-daily data were averaged depends on the specific data provider and the availability of daily measurements. For example, within the same provider, some time series are available as daily measurements, while others include hourly or sub-hourly data. These were daily averaged for air temperature (0-24h clearly), while precipitation totals are computed to the period defined by each provider. Unfortunately, it is not feasible to provide detailed information for each case in the manuscript due to the variability in data sources. However, we clarified this variability in the text (L 180-182).*

**L.178 i understand that the thresholds indicated in Table 2 are used to detect possible problems, which are then manually checked. But how data have been finally considered as "definitely erroneous"? It is important for end users of the dataset to know precisely which criteria have been applied to finally reject a data.**

**It could be, and this should be explained, that these erroneous values were very easy to manually spot out of the adjacent days. If, instead, the detection of erroneous days was semi-automatic, then thresholds are important. I could expect that Tmin could change by more than 20ºC over two days in winter. Concerning precipitation limit, the discussion L.190-192 on extreme precipitation is not clear to me. Daily totals above 500mm are rare, but have been consistently measured (e.g., in the SE part of French Massif Central). And daily, or 24h, totals close to 1000mm can be found in the meteorological archives within the EEAR.**

*Reply: The thresholds used are widely documented in the related references and used by several studies. They are not perfect, and we can experience extreme conditions that, despite being flagged as erroneous, are not in reality. This is the reason why we performed a manual validation of quality control results, that avoid the removal of valid observations. We know that values above 500 mm of precipitation can be found, and we clearly explained it in the text (L 207-208). Data are considered definitely erroneous based on available information in meteorological archives and whether different quality checks agree (e.g. if temperature increases more than 20°C in the next day, it is above 50°C and flagged as outlier, it is clearly a wrong observation). However, note that after this step, the values are further checked by a*

*fully automatic spatial quality control. We clarified how values are set as erroneous by manually checking (L195-196).*

**L.178 Table 2: P > 9 times P95 and P > 5 times P95**

*Reply: As in the comment above (L.125), we replaced "TP" with "P".*

**L.347-349 i do not understand the test here: homogenised series have been used to test 2 procedures (Exp1-2), why did the authors expect to detect heterogeneities in them?**

*Reply: Each homogenization algorithm performs differently, as demonstrated by the Multi-Test Project (Guijarro et al., 2023, see [https://www.climatol.eu/MULTITEST/](https://www.climatol.eu/MULTITEST/)). Consequently, we expected that Exp 1, requiring agreement among all methods, would be more restrictive compared to Exp 2, based on agreement for 2 out of 3 methods. This difference in skill is evident in the results summarized in Lines 372-374. We clarified this in the text for better understanding (L369-370).*

**L.352 Homogenisation procedure: how many years before and after each breakpoint have been used to calculate the quantiles? (not clear in Squintu et al whether 5 or 20 years)**

*Reply: The quantiles calculation does not rely on a fixed-length window. As described in Squintu et al. (2020), the minimum number of years used is 5, and the maximum is 20, depending on data availability around the breakpoint. This process is well illustrated in detail in Figure 2 of Squintu et al. (2020).*

**L.363-365 Any possibility to couple the breakpoint detection and homogenisation for all temperature variables (Tn, Tx, Tm)? Would that improve the correction?**

*Reply: It is possible to perform breakpoint detection and homogenization coupling temperature variables but not all homogenization methods can handle it. For example, we know that some methods such as Climatol or MASH permit it, while others such as ACMANT, HOMER and RH Test do not. Coupled homogenization improves physical consistency among Tmin, Tmean and Tmax without the need to check it after adjustments (see lines 391-398). However, note that in some cases (e.g. shading), biases may affect only one temperature variable and thus need tailored corrections. Moreover, coupled homogenization requires more computational resources and thus it might be complex to apply to large datasets such as the one presented in this study (see lines 311-313). In our case, we combined two homogenization methods that do not allow coupling and one that does, making it impossible to perform a coupled homogenization with this configuration.*

**L.405 "varied" > "variable"?**

*Reply: We agree that "variable" is more precise in this context, and we have updated the manuscript accordingly.*

**L.434 "Most of them consist of at least 60 years" this does not seem consistent with Figs2 and Table A1 which show that most are shorter than 30 years?**

*Reply: We corrected in the manuscript, specifying the exact percentages of available data (L480-483).*

**L.441 "newly digitized data" i do not think this information is given elsewhere, although important; it should be underlined at the beginning**

*Reply: We agree that this is an important detail and have underlined it earlier in the manuscript, specifically in the data collection section (L133).*

**L. 451 "QC" "improved the overall accuracy" : this is expected, of course, but this is not tested/shown?**

*Reply: The improvement of the overall accuracy is shown in Table 4, summarizing the main statistics of quality control outcomes for each data provider and the dataset as a whole. The removal of 0.6% (T) and 3.4% (P) quality issues, while maintaining a marginal effect on the amount of valid observations, is a clear index of improved dataset accuracy. We revised the text (L.417-419) to better clarify this concept.*

**L.462 "was used as a basis to carry out an extended analysis" but not described in this manuscript > make it clear**

*Reply: We clarified this point in the manuscript by explicitly mentioning that the extended analysis is presented in a subsequent paper, now under preparation.*

**L. 463 "a high-resolution interpolated version" i.e. a gridded version?**

*Reply: Yes, we clarified this point in the manuscript. Thank you for the suggestion.*

**L.478 "Values below 0.1 mm" per day? (homogenisation procedure is based on monthly averages)**

*Reply: Yes, the break detection is based on monthly averages. However, the quantile matching method applies corrections derived from monthly averages to daily values (see L 381-382). To ensure consistency and avoid unrealistic results, such as negative precipitation amounts or dry days converted into wet days, these corrections are not applied when daily precipitation is below 0.1 mm. We clarified this concept in the main manuscript (L 543-545).*

**Table A1 should be central in the manuscript since it gives the overall structure of the dataset**

*Reply: In our opinion moving Table A1 to the main discussion would overly condense the presentation of homogenization results without providing additional insights beyond the information already included (see comment 3.2). Indeed, Table A1 primarily summarizes the results of the homogenization process, while the overall structure of the dataset is better represented by the various plots in the data collection section.*

---

## Author Response (AR2)

Dear Editor,

Thank you for accepting the revised version of this manuscript for publication in ESSD. We agree with you and the reviewers that a comparison of this dataset with other related observational products is an important step. We planned to extensively undertake it in a dedicated paper. The manuscript has been revised addressing all technical correction suggested by reviewer#1. Please find below our point-by-point replies (in italics) to the reviewer's comments (in bold).

With our warmest regards,

Giulio Bongiovanni on behalf of all co-authors.

**L.123 space missing: winds(Serafin**

*Reply: Done*

**L.137 "Fig. 2a shows the distribution of station availability as a function of the minimum length of their time series, grouped in 10-year increments"**

**Suggestion: Fig. 2a shows the cumulative distribution of station length (with a 10-yr resolution).**

*Reply: Done*

**L.142 "Fig. 2b instead shows the overall distribution of available time series"**

**Suggestion: Fig. 2b is the histogram of available time series (with a 1-yr resolution).**

*Reply: Done*

**L. 147 "challenging to show in a clear way given the high amount of stations"**

**>> The 'heatmap' provided in the response to our review is very clear and impressive; why not showing it in an Annex?**

*Reply: The heatmap provided in the response to the review has been added in appendix A*

**Figure 2. Distribution of stations by time series length (a) and time (b). Coloured lines identify each variable: mean (in red), maximum and minimum (in green) air temperature,**

and precipitation (in light blue). In a) the minimum time series length, grouped into 10-year increments, up to 2020 is represented. For example, at 30 years, the diagram shows the number of stations with at least 30 years of data, irrespective of the specific time span covered.

Suggestion: Figure 2. Cumulative distribution of time series length with a 10-yr resolution (a), and histogram of time series (b). Coloured lines identify each variable: mean (in red), maximum and minimum (in green) air temperature, and precipitation (in light blue). (In (a), the largest values are the total number of time series, the values for a length of 30 yrs are the number of stations with at least 30 years of data, etc.)

*Reply: Done*

L.165 Suggestion: The higher spatial resolution of the rain-gauge network is evident,

*Reply: Done*

Table 2. The multiplication sign is still missing in the lower part : P > 9 x p95, P > 5 x p95

*Reply: The multiplication sign has been added in the formula of the lower part of Table 2.*

L.192 "avoiding the removal of valid observations" :

L.229 "data flagged with warning flags are automatically replaced with missing values"

L.264 "flagged as missing data"

>> the result of the QC is not yet clear to me, is it to flag data or to remove data?

*Reply: QC flag data with warnings and flagged data have been replaced with missing values. No data has been explicitly removed. We deleted the sentence "avoiding the removal of valid observations" to avoid misinterpretations.*

L.328 space missing: org/)T

*Reply: Done*

L.487 "reliability depends on the availability of heated rain gauges"

**> providing that the heater works! Heating gauges is only one aspect of reducing precipitation measurement bias; reducing wind exposition eg with wind deflector, is another, etc.**

*Reply:* We integrated the sentence mentioning the aspect related to wind exposition in addition to the heater, specifying that these are the two most important among possible factors affecting precipitation measurement bias.

**L.523 "as raw, quality checked and homogenized time series"**

**> as 3 separate time series?**

*Reply:* Yes, we further specified this.